# Finite-time generalized synchronization of non-identical fractional order chaotic systems and its application in speech secure communication

**Jianxiang Yang** [1], **Jianbin Xiong**[1], **Jian Cen**[1,2]*, **Wei He**[1]

**1** School of Automation, Guangdong Polytechnic Normal University, Guangzhou, China, **2** Guangzhou Key Laboratory of Intelligent Building Equipment Information Integration and Control, Guangzhou, China

* 986412676@qq.com

**Data Availability Statement:** All relevant data are within the paper and its Supporting Information files.

## Abstract

This paper focuses on the finite-time generalized synchronization problem of non-identical fractional order chaotic (or hyper-chaotic) systems by a designing adaptive sliding mode controller and its application to secure communication. The effects of both disturbances and model uncertainties are taken into account. A novel fractional order integral sliding mode surface is designed and its stability to the origin is proved in a given finite time. By the aid of the fractional Lyapunov stability theory, a robust controller with adaptive update laws is proposed and its finite-time stability for generalized synchronization between two non-identical fractional-order chaotic systems in the presence of model uncertainties and external disturbances is derived. Numerical simulations are provided to demonstrate the effectiveness and robustness of the presented approach. All simulation results obtained are in good agreement with the theoretical analysis. According to the proposed generalized finite-time synchronization criterion, a novel speech cryptosystem is proposed to send or share voice messages privately via secure channel. Security and performance analyses are given to show the practical effect of the proposed theories.

## 1 Introduction

Chaos synchronization between two identical or non-identical systems is a fascinating problem in nonlinear sciences. Since the pioneering work of Pecora and Carroll [1], the synchronization problem has been widely studied in various fields of science and engineering such as finance system, mechanical systems, power system, encryption, secure communications, and etc. In recent years, the synchronization problem between fractional-order chaotic systems has raised great attentions for its potential applications, especially in cryptography and secure communication. At present, there are various types of fractional-order synchronization, for example, complete synchronization [2], lag synchronization [3], anti-synchronization [4], impulsive synchronization [5], projective synchronization [6], and generalized synchronization [7]. Among all kinds of synchronizations, the generalized synchronization [8] between

**Funding:** This study was supported in part by the National Natural Science Foundation of China under Grant no. 62073090, 61473331, in part by the Guangzhou Key Laboratory of Intelligent Building Equipment Information Integration and Control under Grant no. 202002010003, in part by the Natural Science Foundation of Guangdong Province of China under Grant no.2019A1515010700, in part by the Key (natural) Project of Guangdong Provincial under Grant no. 2019KZDXM020,2019KZDZX1004, 2019KZDZX1042, in part by the Introduction of Talents Project of Guangdong Polytechnic Normal University of China under Grant no. 991641277, 991512203, in part by Guangdong Climbing Project no. pdjh2020b0345, in part by Special projects in key areas of ordinary colleges and universities in Guangdong Province no. 2020ZDZX2014,Intelligent Agricultural Engineering Technology Research Centre of Guangdong University Grant no.ZHNY1905, in part by the Innovation Team Project of Ordinary University of Guangdong Province under Grant no.2020KCXTD017, in part by the Guangdong Special Project in Key Field of Artificial Intelligence for Ordinary University under Grant no. 2019KZDZX1004, in part by the Guangzhou Yuexiu District Science and Technology Plan Major Project under Grant no. 2019-GX-010. The funders had no role in study design, data collection and analysis, decision to publish, or preparation of the manuscript.

**Competing interests:** The authors have declared that no competing interests exist.

the drive system and the response system characterized by two optional functions could obtain desired types in practice applications. Particularly, it can be used to extend the coexistence of different synchronization types. Very recently, the generalized synchronization between two dynamical systems with different dimensions has been studied in [9–11]. Ouannas et al. [12] explored the coexistence of different synchronization types of fractional-order chaotic systems with different dimensions. Wang et al. [13] reported the synchronization between non-identical fractional-order chaotic and hyper-chaotic systems with different orders. Golmankhaneh et al. [14] reported the study of synchronization in non-identical fractional-order chaotic systems.

Note that, the synchronization in non-identical fractional-order chaotic systems can obtain more flexible response mechanism. In practical applications, the mismatched parameters and the uncertainties of master system and slave system are unavoidable. Thus, it is essential to consider and analyze the uncertainties and disturbances. Furthermore, lots of scholars have studied the synchronization method for chaotic systems with different uncertainties, such as the linear feedback method [9], adaptive-feedback scheme [10], adaptive fuzzy approach [11], back-stepping strategy [15], sliding mode control(SMC) [16,17] and adaptive sliding mode control. Amongst these methods, SMC achieved a fast convergence performance and high robustness against the system uncertainties and external disturbances. The generalized robust synchronization approach for mismatched fractional order dynamical systems with different dimensions via sliding mode control was investigated in [13,18]. As we all known, SMC usually assumes the upper bound of the system uncertainties in advance. Nevertheless, in practice, the upper bound may not be exactly known because of the complexity of uncertainties. Therefore, an adaptive mechanism combining the superiority of SMC has been proposed to estimate the unknown bounds of the system uncertainties. In [19], the authors investigated sliding mode synchronization of multiple uncoupled integer order chaotic systems with uncertainties and disturbances, and more general cases were not established on multiple coupled chaotic systems with unknown parameters and disturbances. Further, Chen et al.[20] proposed adaptive sliding mode synchronization for multiple chaotic systems with unknown parameters and disturbances, and the appropriate adaptive laws were given to estimate unknown parameters. In addition, the adaptive sliding mode synchronization of fractional order chaotic systems have been also discussed by researchers [21,22].

However, most of previous studies focused only on the asymptotical synchronization [23]. In practice, it is more valuable and preponderant to study the synchronization in a given finite time other than that in an unpredictable infinite time. For the finite-time stability methods [24], Cai et al. [25] studied the generalized synchronization in finite time among chaotic systems with different order. Further, Zhao et al. [26] investigated the generalized synchronization of integer-order coupled chaotic systems within finite time. Zhang et al. [27]also implemented global synchronization of two integer-order chaotic systems with different dimensions. Chen et al. [28] investigated finite-time multi-switching synchronization of multiple uncertain complex chaotic systems with network transmission mode, and the unknown parameters and disturbances were considered. Furthermore, based on fractional-order the finite-time stability methods, Wu et al. [29] investigated the global synchronization in finite time between non-identical fractional order neural networks (FNNs). With respect to finite-time synchronization, some more results have also been found in [30,31]. Whereas, most of those results focused on the synchronization of identical (non-identical) fractional order dynamical systems either the generalized synchronization in infinite time or without considering the effects of the system uncertainties. It is known that the application of synchronization in secure communication process, chaos-based cryptography can offer a fast and secure way for information protection [32]. The minimum synchronization error and time can be

required to recover or send the encoded message. Besides, for in the chaotic masking [33,34], digital sound encryption techniques in fractional order chaotic systems with a higher level of security are desired, and it can give a powerful solution along with algorithms. Then, it is necessary to study finite-time chaos synchronization problems of fractional order dynamical systems with uncertainties. More importantly, considering the actual situation [34], state variables and uncertainties in the error system have crucial influence on encryption and decryption process of the message signal. Therefore, how to design a controller, which can efficiently reduce the synchronization errors in finite-time and to maximally ensure information transmission security, is a significant and challenging topic.

Inspired by aforementioned previous works, the contributions of this paper can be summarized on four aspects.

1. Based on the definition of the generalized synchronization, the synchronization schemes of two non-identical fractional order chaotic systems are proposed to achieve finite-time generalized synchronization with considering the uncertainties and external disturbances.

2. To study generalized synchronization, a novel fractional order integral sliding mode surface is designed and its stability to the origin is proved in a given finite time.

3. According to the fractional Lyapunov stability theory, an appropriate sliding mode controller with adaptive update laws is proposed under external disturbances and model uncertainties, and the stability conditions for achieving the generalized synchronization are explicitly derived in finite time.

4. Numerical simulation results further highlight the validity, the novelty and applicability of the proposed approach for non-identical fractional order chaotic systems. For the application of secure communication process, a new speech encryption system is introduced to share voice messages secretly via secure channel in terms of the proposed synchronization criterion. Meanwhile, the security of the proposed theories is also analyzed and discussed.

The remainder framework of this study is arranged as follows. In Section 2, the preliminary definition and lemma knowledge necessary are reviewed throughout the paper. The generalized synchronization scheme is introduced in Section 3. Then, numerical simulations are carried out to highlight the effectiveness and applicability of the proposed approach in Section 4. The application in speech secure communication is described in Section 5. Finally, the conclusion is given in Section 6.

## 2. Definitions

In this section, some remarkable definitions of fractional calculus and some helpful lemmas are recalled in the following.

**Definition 1** [35] The $\alpha$th-order Caputo fractional integral of a function $f(t)$ is described by

$$_{t_0}^{C}I_t^\alpha f(t) = \frac{1}{\Gamma(\alpha)}\int_{t_0}^{t}(t-\tau)^{\alpha-1}f(\tau)d\tau, \ \alpha > 0 \tag{1}$$

Where, $1 > \alpha > 0$, $\Gamma(\cdot)$ denotes the gamma function and $\alpha\Gamma(\alpha) = \Gamma(\alpha+1)$.

**Definition 2** [35] The $\alpha$th-order Caputo fractional derivative of a function $f(t)$ is defined as:

$$_{t_0}^{C}D_t^\alpha f(t) = \begin{cases} \dfrac{1}{\Gamma(m-\alpha)}\displaystyle\int_{t_0}^{t}\dfrac{f^{(m)}(\tau)}{(t-\tau)^{\alpha-m+1}}d\tau, m-1 < \alpha < m \\ \dfrac{d^m f(t)}{dt^m}, \alpha = m \end{cases} \tag{2}$$

Where, $1>\alpha>0$ and m is the smallest integer number.

**Lemma 1** [35] When the fractional-order derivative ${}_{t_0}^{C}D_t^{\alpha}x(t)$ is integrable, let $\Omega = [a,b]$ be an interval on the real axis $\mathbb{R}$, and let $n = [\alpha]+1$ for $\alpha \notin N$ or $n = \alpha$ for $\alpha \in N$. If $x(t) \in C^n[a,b]$, one can obtain:

$$
{}_{a}^{C}I_{ta}^{\alpha C}D_t^{\alpha}x(t) = x(t) - \sum_{k=0}^{n-1} \frac{x^{(k)}(a)}{k!}(t-a)^k, n-1 < \alpha \leq n \tag{3}
$$

Especially, if $0<\alpha\leq 1$ and $x(t)\in C^1[a,b]$, then ${}_{a}^{C}I_{ta}^{\alpha C}D_t^{\alpha}x(t) = x(t) - x(a)$.

**Lemma 2** [35] Assume $\alpha\in(0,1)$, $p\in R$, then

$$
{}_{t_0}^{C}D_t^{\alpha}x^p(t) = \frac{\Gamma(1+p)}{\Gamma(1+p-\alpha)}x^{p-\alpha}(t){}_{t_0}^{C}D_t^{\alpha}x(t) \tag{4}
$$

**Lemma 3** [29,36] Suppose $\alpha\in(0,1)$, and $x(t)$ denotes a continuous and differentiable function, then it satisfies the following inequality

$$
\begin{cases}
{}_{t_0}^{C}D_t^{\alpha}|x(t)| \leq sign(x(t)){}_{t_0}^{C}D_t^{\alpha}x(t) \\
\frac{1}{2}{}_{t_0}^{C}D_t^{\alpha}x^2(t) \leq x(t){}_{t_0}^{C}D_t^{\alpha}x(t)
\end{cases} \tag{5}
$$

**Lemma 4** [27] If $d_i\in\mathbb{R}$, $i = 1,2\cdots n$ and $\xi\in(0,1)$ are arbitrary real numbers, the following inequalities satisfy:

$$
|d_1|^{\xi} + |d_2|^{\xi} + \cdots + |d_n|^{\xi} \geq (|d_1| + |d_2| + \cdots + |d_n|)^{\xi} \tag{6}
$$

**Lemma 5** [37] Consider the following n-dimensional fractional-order dynamical system

$$
\begin{cases}
{}_{t_0}^{C}D_t^{\alpha}x(t) = f(t,x) \\
x(0) = x_0
\end{cases} \tag{7}
$$

Where, $\alpha\in(0,1)$, $x(t)\in\mathbb{R}^n$ is the system state and $f:[0,\infty)\times\mathbb{R}^n\to\mathbb{R}^n$ is a continuous nonlinear function. Assume that there exists a continuously differential Lyapunov function $V(t,x(t))$ and strictly increasing class-K functions $b_1$, $b_2$ and $b_3$ satisfying

$$
\begin{cases}
b_1(\|x\|) \leq V(t,x(t)) \leq b_2(\|x\|) \\
{}_{t_0}^{C}D_t^{\beta}V(t,x(t)) \leq -b_3(\|x\|)
\end{cases} \tag{8}
$$

Where, $\beta\in(0,1)$, Then the equilibrium point $x = 0$ of the fractional-order system (7) is asymptotically stable.

**Remark 1** For simplicity, the Caputo fractional calculus of order $\alpha$ as ${}_{t_0}^{C}D_t^{\alpha}$ and ${}_{t_0}^{C}I_t^{\alpha}$ are substituted for $D^{\alpha}$ and $I^{\alpha}$, respectively.

## 3. The generalized synchronization scheme

In this section, the main goal is to develop the generalized synchronization between two non-identical fractional order chaotic/hyper-chaotic systems in finite-time, which play an important role to acquire the main results via applying sliding mode technique.

Consider the following n-dimensional fractional order master system

$$D^\alpha x = F(x) + \Delta f(x) + d^f(t) \tag{9}$$

Where, $0 < \alpha \leq 1$ is the fractional order of the system, $x = [x_1, x_2, \cdots, x_n]^T$ is the system state vector; $f(x) = [\Delta f_1(x_1), \Delta f_2(x_2), \cdots, \Delta f_n(x_n)]^T \in \mathbb{R}^n$, $d^f(t) = [d_1^f(t), d_2^f(t), \cdots, d_n^f(t)]^T \in \mathbb{R}^n$ denote unknown model uncertainties and external disturbances, respectively. $F(x) = [F_1(x), F_2(x), \cdots F_n(x)]^T$ is a nonlinear function.

Consider the corresponding $m$- dimensional fractional order slave system

$$D^\beta y = G(y) + \Delta g(y) + d^g(t) + U(t) \tag{10}$$

Where, $0 < \beta \leq 1$ is the fractional order of the system, $y = [y_1, y_2, \cdots, y_m]^T$ is the system state vector; $\Delta g(y) = [\Delta g_1(y_1), \Delta g_2(y_2), \cdots, \Delta g_m(y_m)]^T$, $d^g(t) = [d_1^g(t), d_2^g(t), \ldots, d_m^g(t)]^T$ denote unknown model uncertainties and disturbances or perturbations, respectively. And $G(y) = [G_1(y), G_2(y), \cdots, G_m(y)]^T$ is a nonlinear function; the control input is $U(t) = [u_1(t), u_2(t), \cdots, u_m(t)]^T$.

Now, the definition of generalized synchronization between fractional-order chaotic systems is given in the following expression.

**Definition 3** Consider the above systems (9) and (10) with different initial values denoted by $x_0$ and $y_0$. Assume that there exist an open neighborhood $\Theta \subset \mathbb{R}^r$ of the origin, two continuously differentiable functions $\phi: \mathbb{R}^n \rightarrow \mathbb{R}^r$ and $\varphi: \mathbb{R}^m \rightarrow \mathbb{R}^r$, i.e., $e_0 = \varphi(y_0) - \phi(x_0) \in \Theta$, and a constant $T = T(e(0)) \in (0, \infty)$, one can get

$$lim_{t \rightarrow T} \|e(t)\| = lim_{t \rightarrow T} \|\varphi(y) - \phi(x)\| = 0 \tag{11}$$

Where, $e(t) \in \mathbb{R}^r$ denotes the synchronization error of the master system (15) and slave system (16). Then, $\|e(t)\| \equiv 0$, $t \geq T$, that is, the synchronization error can be achieved to be zero within a finite time.

**Remark 2** Fractional-order chaotic systems with same dimensions when $m = n$, that is $\phi(x(t)) = x(t)$, $\varphi(y(t)) = y(t)$, if the synchronization error is $e(t) = y(t) - x(t)$, it can be transformed into globally complete synchronization; if the synchronization error is $e(t) = x(t) - y(t)$, it can accomplish globally anti-synchronization; if the synchronization error is $e(t) = y(t) - px(t)$, and $p$ is the projective coefficient, it can become globally projective synchronization; if the synchronization error is $e(t) = x(t)$, the above Eq (11) can be inferred as the following form $lim_{t \rightarrow T} \|x(t)\| = 0$, it will be transformed into the stabilization of the master system. Obviously, these are special cases of our proposed methods. Zhang et al. [26] proposed the finite-time synchronization of the error system $e(t) = \varphi(y) - \phi(x)$, but the author did not consider disturbances and model uncertainties, and the error system should be also integer-order system. Nevertheless, this idea is appreciated.

Defining $e(t) = \varphi(y) - \phi(x)$, from system (9) and system (10), in order to introduce conveniently, the control signals $U(t)$ are considered as $U'(t)$ and $U''(t)$, i.e., $U(t) = U'(t) + U''(t)$. The compensation controller $U''(t)$ can be preset as $U''(t) = J_\varphi^{-1} J_\phi (D^\beta x - D^\alpha x)$, and the separated controller $U'(t)$ will be designed later. Then, we have

$$D^\beta e_i(t) = D^\beta [\varphi(y) - \phi(x)] = J_\varphi(y) D^\beta y - J_\phi(x) D^\beta x$$

$$= J_\varphi(y) \begin{pmatrix} G(y) + \Delta g(y) \\ + d^g(t) + U'(t) \end{pmatrix} - J_\phi(x)(F(x) + \Delta f(x) + d^f(t)) \qquad i = 1, 2, \ldots, r. \tag{12}$$

Where, $J_\phi(x)$ and $J_\varphi(y)$ are the Jacobin matrices of the functions $\phi(x)$ and $\varphi(y)$, respectively, i.e.

$$J_Q(x) = \begin{bmatrix} \dfrac{\partial Q_1(x)}{\partial x_1} & \dfrac{\partial Q_1(x)}{\partial x_2} & \cdots & \dfrac{\partial Q_1(x)}{\partial x_n} \\ \dfrac{\partial Q_2(x)}{\partial x_1} & \dfrac{\partial Q_2(x)}{\partial x_2} & \cdots & \dfrac{\partial Q_2(x)}{\partial x_n} \\ \vdots & \vdots & \ddots & \vdots \\ \dfrac{\partial Q_r(x)}{\partial x_1} & \dfrac{\partial Q_r(x)}{\partial x_2} & \cdots & \dfrac{\partial Q_r(x)}{\partial x_n} \end{bmatrix}, J_P(y) = \begin{bmatrix} \dfrac{\partial P_1(y)}{\partial y_1} & \dfrac{\partial P_1(y)}{\partial y_2} & \cdots & \dfrac{\partial P_1(y)}{\partial y_m} \\ \dfrac{\partial P_2(y)}{\partial y_1} & \dfrac{\partial P_2(y)}{\partial y_2} & \cdots & \dfrac{\partial P_2(y)}{\partial y_m} \\ \vdots & \vdots & \ddots & \vdots \\ \dfrac{\partial P_r(y)}{\partial y_1} & \dfrac{\partial P_r(y)}{\partial y_2} & \cdots & \dfrac{\partial P_r(y)}{\partial y_m} \end{bmatrix}$$

**Remark 3** $r \leq min\{m, n\}$, the Jacobin matrix $J_\varphi$ is row full-rank, it is well known that $J_\varphi$ is a square matrix, the inverse matrix $J_\varphi^{-1}$ exists. In terms of the generalized inverse matrix definition, the right inverse matrix $J_{\varphi R}^{-1}$ exists when it is not a square matrix. To simplify the symbol, $J_\varphi^{-1}$ denotes the inverse or right inverse matrix of $J_\varphi$ in this study.

**Assumption 1** It is assumed that disturbances $d^f(t)$ and model uncertainties $\Delta f(x)$ of the master system (15) are all bounded, there exist unknown positive constants $\gamma_i$, i.e., $|J_\phi(\Delta f(x) + d^f(t))| \leq \gamma_i, i = 1,2,\ldots,r$.

**Assumption 2** The disturbances $d^g(t)$ and model uncertainties $\Delta g(x)$ of the slave system (10) are assumed to be all bounded, there also exist unknown positive constants $\varepsilon_i$, i.e., $|J_\varphi(\Delta g(y)+d^g(t))| \leq \varepsilon_i, i = 1,2,\ldots,r$.

**Remark 4** The objective of this study can be formulated as designing an appropriate control law $U^r(t)$ for any different dimensional systems (9) and (10) with disturbances and model uncertainties, the finite-time stability for the error system (12) can be accomplished in the light of Definition 3.

To further study generalized synchronization of two chaotic systems (9) and (10) with different dimensions, it can be transformed into the globally stability of equilibrium point for error system (12). Here, a sliding mode technique will be used to solve the generalized synchronization problem. In general, the design process of sliding mode control includes the following two major steps. The first step is to determine a suitable sliding surface with some required system dynamic characteristics. Second, the appropriate sliding mode control laws are arranged to guarantee the state trajectories onto the sliding surface and subsequently stay on it forever. Therefore, a novel fractional integral siding surface is constructed as follows:

$$s_i(t) = e_i(t) + k_1 I^\beta |e_i(t)|^\delta sign(e_i(t)), i = 1, 2, \ldots, r. \tag{13}$$

Where $s_i(t) = [s_1, s_2, \ldots, s_r]^T \in \mathbb{R}^r$ is the sliding surface, $e_i(t) = [e_1, e_2, \ldots e_r] \in \mathbb{R}^r$ is the synchronization error state, $0 < \delta < \beta < 1$, $k_1 > 0$ is the gain coefficient.

Based on the sliding mode control strategy, the sliding surface and its derivative should satisfy: $s_i(t) = 0$ and $\dot{s}_i(t) = 0$. We know $\dot{s}_i(t) = D^{1-\beta}D^\beta s_i(t)$, then, $\dot{s}_i(t) = 0$ implies $D^\beta s_i(t) = 0$. Therefore, from (13), one has

$$D^\beta s_i(t) = D^\beta e_i(t) + k_1 |e_i(t)|^\delta sign(e_i(t)) = 0 \rightarrow D^\beta e_i(t) = -k_1 |e_i(t)|^\delta sign(e_i(t)) \tag{14}$$

Based on Lemma 5, the following finite time convergence theorem of the fractional terminal sliding surface (13) is analytically proved.

**Theorem 1** Consider the sliding mode dynamics (14). The error system will be global asymptotically stable and converge to the equilibrium $e(t) = 0$ within finite time upper

 

bounded by:

$$T_1^* = t_0 + \left( \|e(t_0)\|_1^{\beta-\delta} \frac{\Gamma(1-\delta)\Gamma(1+\beta)}{k_1\Gamma(1+\beta-\delta)} \right)^{\frac{1}{\beta}}$$

**Proof** Select the following Lyapunov function candidate:

$$V_1(t) = \|e(t)\|_1 = \sum_{i=1}^r |e_i(t)| \tag{15}$$

By applying **Lemma 3**, one has

$$D^\beta V_1(t) \leq \sum_{i=1}^r sign(e_i(t))D^\beta e_i(t) \tag{16}$$

Substituting $D^\beta e_i(t)$, $i = 1,2,\ldots,r$ from (14) into (16), and $sign(e_i)\times sign(e_i) = 1$, one obtains

$$D^\beta V_1(t) \leq - \sum_{i=1}^r sign(e_i(t))$$
$$k_1 |e_i(t)|^\delta sign(e_i(t)) = -k_1 \sum_{i=1}^r |e_i(t)|^\delta \tag{17}$$

Using **Lemma 4** the following inequality $\sum_{i=1}^r |e_i|^\delta \geq (\sum_{i=1}^r |e_i|)^\delta$, one gets

$$D^\beta V_1(t) \leq -\sum_{i=1}^r k_1 |e_i(t)|^\delta \leq -k_1 V_1^\delta(t) \tag{18}$$

Based on **Lemma 5**, the dynamic error $e_i(t)$, $i = 1,2,\ldots,r$ will converge to zero asymptotically. In terms of Lemma 2, one has

$$D^\beta V_1(t) = \frac{\Gamma(1-\delta)}{\Gamma(1+\beta-\delta)} V_1^\delta(t)D^\beta V_1^{\beta-\delta}(t) \leq -k_1 V_1^\delta(t) \tag{19}$$

From (19), it can be easy to derive the following form:

$$D^\beta V_1^{\beta-\delta}(t) \leq -k_1 \frac{\Gamma(1+\beta-\delta)}{\Gamma(1-\delta)} \tag{20}$$

Taking fractional-order integral of (20) from $t_0$ to t by Lemma 1, one obtains

$$V_1^{\beta-\delta}(t) - V_1^{\beta-\delta}(t_0) \leq I^\beta \left( -\frac{k_1\Gamma(1+\beta-\delta)}{\Gamma(1-\delta)} \right) \tag{21}$$

According to **Definition 1**, one gets

$$I^\beta \left( -\frac{k_1\Gamma(1+\beta-\delta)}{\Gamma(1-\delta)} \right) = \frac{-k_1\Gamma(1+\beta-\delta)}{\Gamma(1-\delta)} \frac{1}{\Gamma(\beta)} \int_0^t (t-\tau)^{\beta-1} d\tau$$
$$= \frac{-k_1\Gamma(1+\beta-\delta)}{\Gamma(1-\delta)\Gamma(\beta)} \frac{(t-t_0)^\beta}{\beta} = \frac{-k_1\Gamma(1+\beta-\delta)(t-t_0)^\beta}{\Gamma(1-\delta)\Gamma(1+\beta)} \tag{22}$$

Combining (21) and (22), one can get

$$V_1^{\beta-\delta}(t) \leq V_1^{\beta-\delta}(t_0) - \frac{k_1\Gamma(1+\beta-\delta)(t-t_0)^\beta}{\Gamma(1-\delta)\Gamma(1+\beta)}, \quad t_0 \leq t \leq T_1^* \tag{23}$$

From (23), one can obtain that $lim_{t\to T_1^*} V_1(t) = 0$, such that $V_1(t) = 0$ for arbitrary $t \geq T_1^*$, and the sliding-mode dynamic error $e_i(t)$, $i = 1,2,\ldots,r$ will converge to zero in finite time, i.e.,

$lim_{t \to T_1^*} e_i(t) = lim_{t \to T_1^*} \|\phi(x) - \varphi(y)\| = 0, \ t \geq T_1^*. T_1^*$ is the upper bound of convergence time, given by $T_1^* = t_0 + \left( \|e(t_0)\|_1^{\beta - \delta} \frac{\Gamma(1-\delta)\Gamma(1+\beta)}{k_1 \Gamma(1+\beta-\delta)} \right)^{\frac{1}{\beta}}$. This completes the proof.

In what follows, in order to satisfy the sliding condition under disturbances and model uncertainties, the adaptive sliding control law is proposed as follows:

$$U'_i(t) = -G(y) + J_\varphi^{-1}(y) \begin{pmatrix} J_\phi(x)F(x) \\ -k_1|e_i(t)|^\delta sign(e_i(t)) \\ -\begin{pmatrix} k_2|s_i(t)|^\sigma \\ +\hat{\varepsilon}_i + \hat{\gamma}_i \end{pmatrix} sign(s_i(t)) \end{pmatrix} \tag{24}$$

Where, $i = 1,2,\ldots,r$, $0<\sigma<\beta<1$, $k_2>0$ is the gain coefficient. $\hat{\varepsilon}_i$ and $\hat{\gamma}_i$ are denoted as estimates of $\varepsilon_i$ and $\gamma_i$, respectively. In this subsection, the adaptive update laws are designed by the following algorithm:

$$\begin{cases} D^\beta \hat{\varepsilon}_i = |s_i| - k_2|\hat{\varepsilon}_i - \varepsilon_i|^\sigma sign(\hat{\varepsilon}_i - \varepsilon_i) \\ D^\beta \hat{\gamma}_i = |s_i| - k_2|\hat{\gamma}_i - \gamma_i|^\sigma sign(\hat{\gamma}_i - \gamma_i) \end{cases} \tag{25}$$

**Theorem 2** Under Assumption 1 and 2, consider the synchronization error system (18) with uncertainties and external disturbances. Based on the control law (24) with the adaptive laws (25), then its trajectories will globally reach the sliding surface $s(t) = 0$ within finite time upper bounded by:

$$T_2^* = t_0 + \left( \frac{\left( \frac{1}{2}s_i^2(0) + \frac{1}{2}(\hat{\varepsilon}_i(0) - \varepsilon_i)^2 + \frac{1}{2}(\hat{\gamma}_i(0) - \gamma_i)^2 \right)^{\beta - \frac{\sigma+1}{2}}}{2^{\frac{\sigma+1}{2}} k_2 \Gamma\left(1 + \beta - \frac{\sigma+1}{2}\right)} \right)^{\frac{1}{\beta}}$$

**Proof** We choose the following Lyapunov function candidate

$$V_2(t) = \frac{1}{2} \sum_{i=1}^r (s_i^2(t) + (\hat{\varepsilon}_i - \varepsilon_i)^2 + (\hat{\gamma}_i - \gamma_i)^2) \tag{26}$$

Based on **Lemma 3**, taking the fractional-order derivative of $V_2(t)$ as follows:

$$D^\beta V_2(t) \leq \sum_{i=1}^r \begin{pmatrix} s_i(t)D^\beta s_i(t) \\ +(\hat{\varepsilon}_i - \varepsilon_i)D^\beta \hat{\varepsilon}_i + (\hat{\gamma}_i - \gamma_i)D^\beta \hat{\gamma}_i \end{pmatrix} \tag{27}$$

Combining (12), (14), (24) and (25), one obtains

$$D^\beta V_2(t) \leq \sum_{i=1}^r \left( s_i(t) \left( \begin{array}{c} J_\varphi(y)(\Delta g(y) + d^g(t)) \\ \\ -J_\phi(x)(\Delta f(x) + d^f(t)) \\ \\ -\left( \begin{array}{c} k_2|s_i(t)|^\sigma \\ \\ +\hat{\varepsilon}_i + \hat{\gamma}_i \end{array} \right) \text{sign}(s_i(t)) \end{array} \right) + (\hat{\varepsilon}_i - \varepsilon_i) \left( \begin{array}{c} |s_i| - k_2 \\ \\ |\hat{\varepsilon}_i - \varepsilon_i|^\sigma \text{sign}(\hat{\varepsilon}_i - \varepsilon_i) \end{array} \right) + (\hat{\gamma}_i - \gamma_i) \left( \begin{array}{c} |s_i| - k_2 \\ \\ |\hat{\gamma}_i - \gamma_i|^\sigma \text{sign}(\hat{\gamma}_i - \gamma_i) \end{array} \right) \right) \tag{28}$$

On the basis of Assumptions 1 and 2, one yields,

$$D^\beta V_2(t) \leq \sum_{i=1}^r \left( \begin{array}{c} |s_i| \left( \begin{array}{c} \gamma_i \\ +\varepsilon_i \end{array} \right) - \left( \begin{array}{c} k_2|s_i(t)|^\sigma \\ +\hat{\varepsilon}_i + \hat{\gamma}_i \end{array} \right) |s_i| \\ +(\hat{\varepsilon}_i - \varepsilon_i)|s_i| + (\hat{\gamma}_i - \gamma_i)|s_i| \\ -k_2(|\hat{\varepsilon}_i - \varepsilon_i|^{\sigma+1} + |\hat{\gamma}_i - \gamma_i|^{\sigma+1}) \end{array} \right) \tag{29}$$

In terms of **Lemma 4**, it is easy to obtain that

$$D^\beta V_2(t) \leq -k_2 \sum_{i=1}^r \left( \begin{array}{c} |s_i(t)|^{\sigma+1} \\ +|\hat{\varepsilon}_i - \varepsilon_i|^{\sigma+1} \\ +|\hat{\gamma}_i - \gamma_i|^{\sigma+1} \end{array} \right) \leq -2^{\frac{\sigma+1}{2}} k_2 \sum_{i=1}^r \left( \begin{array}{c} \left( \frac{1}{2} s_i^2(t) \right)^{\frac{\sigma+1}{2}} \\ +\left( \frac{1}{2}(\hat{\varepsilon}_i - \varepsilon_i)^2 \right)^{\frac{\sigma+1}{2}} \\ +\left( \frac{1}{2}(\hat{\gamma}_i - \gamma_i)^2 \right)^{\frac{\sigma+1}{2}} \end{array} \right) \leq -2^{\frac{\sigma+1}{2}} k_2 V_2^{\frac{\sigma+1}{2}}(t) < 0 \tag{30}$$

On the basis of Lemma 5, the state trajectories of the error system will converge to $s(t) = 0$ asymptotically. By Lemma 2, one has

$$D^\beta V_2(t) = \frac{\Gamma\left(1 - \frac{\sigma+1}{2}\right)}{\Gamma\left(1 + \beta - \frac{\sigma+1}{2}\right)} V_2^{\frac{\sigma+1}{2}}(t) D^\beta V_2^{\beta - \frac{\sigma+1}{2}}(t) \leq -2^{\frac{\sigma+1}{2}} k_2 V_2^{\frac{\sigma+1}{2}}(t) \tag{31}$$

Then, it can be easy to derive the following form:

$$D^{\beta} V_2^{\beta - \frac{\sigma+1}{2}}(t) \leq -2^{\frac{\sigma+1}{2}} k_2 \frac{\Gamma\left(1 + \beta - \frac{\sigma+1}{2}\right)}{\Gamma\left(1 - \frac{\sigma+1}{2}\right)} \tag{32}$$

Taking fractional-order integral of (32) from $t_0$ to $t$ by Lemma 1, one obtains

$$V_2^{\beta - \frac{\sigma+1}{2}}(t) - V_2^{\beta - \frac{\sigma+1}{2}}(t_0) \leq I^{\beta}\left(-2^{\frac{\sigma+1}{2}} k_2 \frac{\Gamma\left(1 + \beta - \frac{\sigma+1}{2}\right)}{\Gamma\left(1 - \frac{\sigma+1}{2}\right)}\right) \tag{33}$$

According to **Definition 1**, one gets

$$\begin{aligned}
I^{\beta}\left(-2^{\frac{\sigma+1}{2}} k_2 \frac{\Gamma\left(1 + \beta - \frac{\sigma+1}{2}\right)}{\Gamma\left(1 - \frac{\sigma+1}{2}\right)}\right) &= -2^{\frac{\sigma+1}{2}} k_2 \frac{\Gamma\left(1 + \beta - \frac{\sigma+1}{2}\right)}{\Gamma\left(1 - \frac{\sigma+1}{2}\right)} \frac{1}{\Gamma(\beta)} \int_0^t (t - \tau)^{\beta-1} d\tau \\
&= -2^{\frac{\sigma+1}{2}} k_2 \frac{\Gamma\left(1 + \beta - \frac{\sigma+1}{2}\right)}{\Gamma\left(1 - \frac{\sigma+1}{2}\right)\Gamma(\beta)} \frac{(t - t_0)^{\beta}}{\beta} = -2^{\frac{\sigma+1}{2}} k_2 \frac{\Gamma\left(1 + \beta - \frac{\sigma+1}{2}\right)(t - t_0)^{\beta}}{\Gamma\left(1 - \frac{\sigma+1}{2}\right)\Gamma(1 + \beta)}
\end{aligned} \tag{34}$$

From (33) and (34), one can obtain

$$V_2^{\beta - \frac{\sigma+1}{2}}(t) \leq V_2^{\beta - \frac{\sigma+1}{2}}(t_0) - 2^{\frac{\sigma+1}{2}} k_2 \frac{\Gamma\left(1 + \beta - \frac{\sigma+1}{2}\right)(t - t_0)^{\beta}}{\Gamma\left(1 - \frac{\sigma+1}{2}\right)\Gamma(1 + \beta)}, t_0 \leq t \leq T_2^*$$

Hence, it implies that the state trajectories of the error system (12) will reach the predefined sliding surface $s(t) = 0$ in a given finite time under the controller (30). One can obtain that

$$\lim_{t \to T_2^*} s_i(t) = 0, \ t \geq T_2^*, \ i = 1, 2 \ldots r \tag{35}$$

Therefore, this completes the proof.

**Remark 5** Theorem 1 has proved that the synchronization error (11) can be achieved to be zero within a finite time, and Theorem 2 has proved that the state trajectories of the error system can converge to zero within a given finite time. Therefore, according to Theorems 1 and 2, the upper bound of total convergence time can be estimated as $T^* < T_1^* + T_2^*$.

**Remark 6** In the proposed synchronization method, all potentialities: dimension, fractional order derivative, identical or non-identical and with or without disturbances and model uncertainties, are included in the dynamical systems. Consequently, the so-called 'generalized synchronization approach' is entirely adequate for synchronizing any dynamical systems in finite time.

**Remark 7** From the perspective of control engineering, the main problems that greatly limits the control performance in practical applications are: a) the model parameters or the upper bound of the dynamical system, b) the presence of uncertainties and external disturbances in the system, c) The feasibility of the control inputs. In the proposed control scheme, it is not necessary to give prior knowledge of the upper bounds. Furthermore, due to the efficiency of the adaptive sliding mode control, it has good robustness against uncertainties and disturbances. It's also noteworthy that for real implementations of the adaptive update laws in Eq (25), the disturbances and model uncertainties need to satisfy the Assumption 1 and 2. Further, the control inputs $U_i'(t)$ are feasible in real applications. Hence, the proposed controller is very suitable for practical applications.

## 4. Numerical simulations

In this section, some numerical simulations are provided to highlight the validity and effectiveness of our proposed methods obtained in the previous section. Two cases are discussed by applying the method to two non-identical fractional order chaotic systems with and without commensurate orders.

In the following, let us consider the hyper-chaotic fractional order Lorenz system (36) as a master system, and chaotic fractional order reverse butterfly-shape system (37) as a slave system. These systems with disturbances and model uncertainties are taken from [18].

Master system:

$$\begin{cases} D^{\alpha}x_1 = 10(x_2 - x_1) + x_4 + \Delta f_1(x_1) + d_1^f(t) \\ D^{\alpha}x_2 = 28x_1 - x_2 - x_1 x_3 + \Delta f_2(x_2) + d_2^f(t) \\ D^{\alpha}x_3 = x_1 x_2 - \left(^8/_3\right)x_3 + \Delta f_3(x_3) + d_3^f(t) \\ D^{\alpha}x_4 = -x_2 x_3 - x_4 + \Delta f_4(x_4) + +d_4^f(t) \end{cases} \tag{36}$$

Slave system:

$$\begin{cases} D^{\beta}y_1 = 10(y_2 - y_1) + \Delta g_1(y_1) + d_1^g(t) + u_1(t) \\ D^{\beta}y_2 = 40y_1 + 16y_1 y_3 + \Delta g_2(y_2) + d_2^g(t) + u_2(t) \\ D^{\beta}y_3 = -y_1 y_2 - \left(^{10}/_4\right)y_3 + \Delta g_3(y_3) + d_3^g(t) + u_3(t) \end{cases} \tag{37}$$

The disturbances and model uncertainties of the systems are chosen as follows:

$$\begin{cases} \Delta f_1(x_1) + d_1^f(t) = -0.15cos(6t)x_1 + 0.25sin(7t) \\ \Delta f_2(x_2) + d_2^f(t) = -0.2cos(2t)x_2 + 0.1sin(3t) \\ \Delta f_3(x_3) + d_3^f(t) = 0.15sin(4t)x_3 - 0.25cos(5t) \\ \Delta f_4(x_4) + d_4^f(t) = -0.2sin(t)x_4 + 0.2cos(2t) \\ \Delta g_1(y_1) + d_1^g(t) = -0.25sin(4t)y_1 + 0.1cos(t) \\ \Delta g_2(y_2) + d_2^g(t) = 0.1cos(2t)y_2 + 0.15cos(3t) \\ \Delta g_3(y_3) + d_3^g(t) = -0.1sin(3t)y_3 + 0.2cos(5t) \end{cases} \tag{38}$$

By using Assumption 1 and 2, one can obtain $\varepsilon_i = (0.1,0.8,0.8)$, $\gamma_i = (0.35,0.6,0.55)(i = 1,2,3)$. The following examples are illustrated by using two varieties of cases for fractional order derivatives: one is $\alpha \neq \beta$ and another one is $\alpha = \beta$.

### 4.1. Non-commensurate order

In our simulation, the fractional derivatives are selected as $\alpha = 0.98$, $\beta = 0.99$ respectively. The initial values of the systems (36) and (37) are given as $(x_1,x_2,x_3,x_4) = (2,-2,4,1)$, $(y_1,y_2,y_3) = (2,-1,1)$. By taking those parameters, the dynamics of master system (36) and slave system (37) without the controller exhibits a chaotic behavior as illustrated in Figs 1 and 2.The corresponding time series of the master and slave systems are shown in Figs 3 and 4.

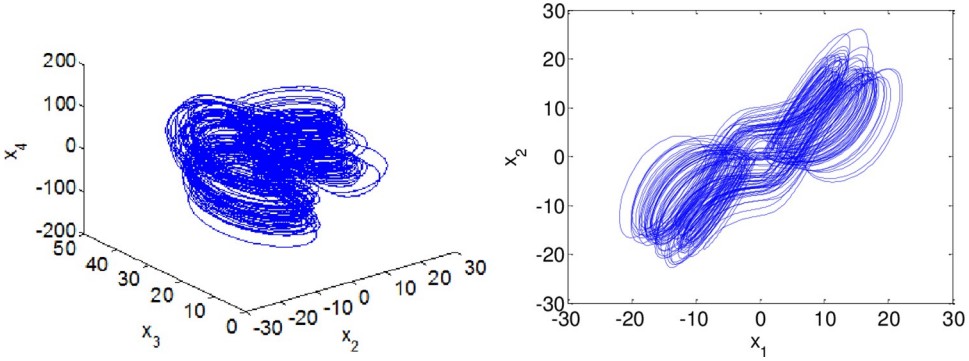

**Fig 1. Phase portraits of master system when α = 0.98.**

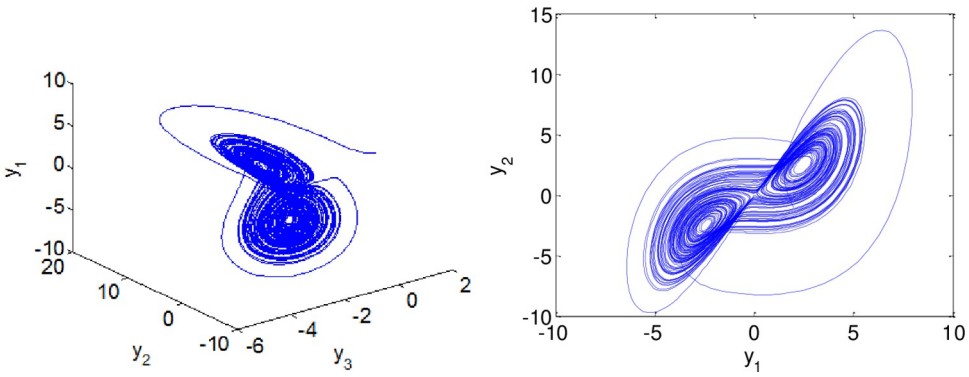

**Fig 2. Phase portraits of the uncontrolled slave system when β = 0.99.**

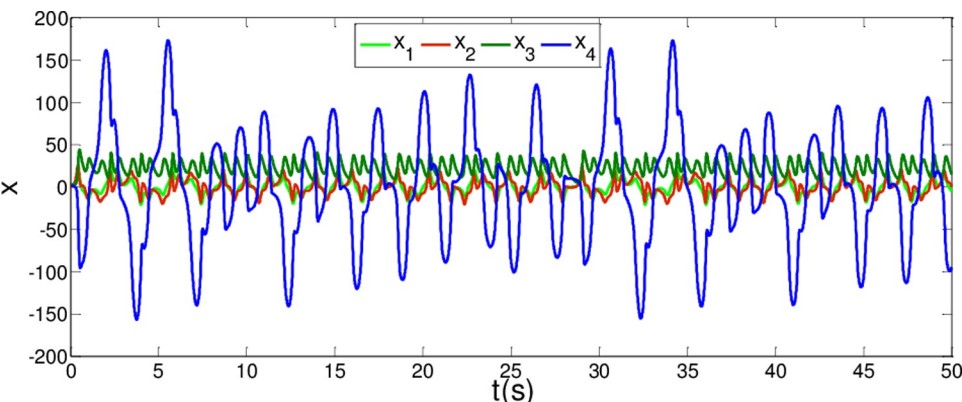

**Fig 3. Time series of master system when α = 0.98.**

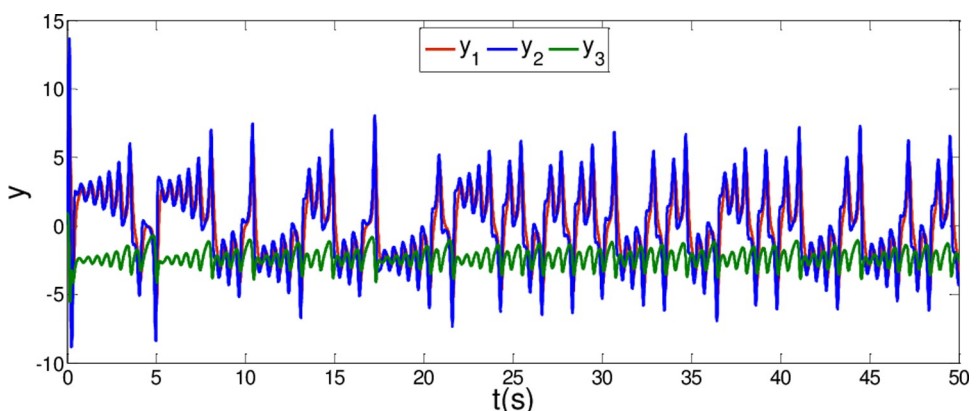

**Fig 4. Time series of the uncontrolled slave system when β = 0.99.**

Assume the continuous differentiable functions of the systems (36) and (37) are

$$\phi(x) = \begin{pmatrix} x_1 - x_2 \\ x_1 + x_3 \\ x_1 + x_4 \end{pmatrix}, \; \varphi(y) = \begin{pmatrix} y_1 \\ y_1 + y_2 \\ y_2 + y_3 \end{pmatrix} \tag{39}$$

Then $e(t) = (e_1(t), e_2(t), e_3(t))^T = \varphi(y) - \phi(x) = \begin{pmatrix} y_1 - x_1 + x_2 \\ y_1 + y_2 - x_1 - x_3 \\ y_2 + y_3 - x_1 - x_4 \end{pmatrix}$. Further, the

corresponding Jacobian matrix and the generalized inverse matrix is obtained

$$J_\phi(x) = \begin{pmatrix} 1 & -1 & 0 & 0 \\ 1 & 0 & 1 & 0 \\ 1 & 0 & 0 & 1 \end{pmatrix}, \; J_\varphi(y) = \begin{pmatrix} 1 & 0 & 0 \\ 1 & 1 & 0 \\ 0 & 1 & 1 \end{pmatrix} \tag{40}$$

By (19) and (27), the sliding surface and control input are provided as follows:

$$s_i(t) = e_i(t) + I^\beta(k_1|e_i(t)|^\delta sign(e_i(t))), \; i = 1, 2, 3 \tag{41}$$

And the control signals $U(t) = U'(t) + U''(t)$, compensation controllers $U''_i(t) = J_\varphi^{-1} J_\phi (D^\beta x - D^\alpha x)$, $i = 1, 2, 3$. From (24), adaptive controllers $U'_i(t)$ are determined by the following

$$
\begin{cases}
U'_1(t) = -10(y_2 - y_1) - 38x_1 + 11x_2 \\
\quad + x_1 x_3 + x_4 - k_1|e_i(t)|^\delta sign(e_1(t)) \\
\quad - (k_2|s_1(t)|^\sigma + \hat{\varepsilon}_1 + \hat{\gamma}_1) sign(s_1(t))) \\
U'_2(t) = -40y_1 - 16y_1 y_3 - 28x_1 - x_2 - x_1 x_3 \\
\quad + x_1 x_2 - \dfrac{8}{3} x_3 + k_1|e_i(t)|^\delta sign(e_1(t)) \\
\quad + (k_2|s_1(t)|^\sigma + \hat{\varepsilon}_1 + \hat{\gamma}_1) sign(s_1(t))) \\
\quad - k_1|e_2(t)|^\delta sign(e_2(t)) \\
\quad - (k_2|s_2(t)|^\sigma + \hat{\varepsilon}_2 + \hat{\gamma}_2) sign(s_2(t))) \\
U'_3(t) = y_1 y_2 + \dfrac{10}{4} y_3 - 38x_1 + 11x_2 + x_1 x_3 \\
\quad - x_1 x_2 - x_2 x_3 + \dfrac{8}{3} x_3 - k_1|e_i(t)|^\delta sign(e_1(t)) \\
\quad - (k_2|s_1(t)|^\sigma + \hat{\varepsilon}_1 + \hat{\gamma}_1) sign(s_1(t))) \\
\quad + k_1|e_2(t)|^\delta sign(e_2(t)) \\
\quad + (k_2|s_2(t)|^\sigma + \hat{\varepsilon}_2 + \hat{\gamma}_2) sign(s_2(t))) \\
\quad - k_1|e_3(t)|^\delta sign(e_3(t) \\
\quad - (k_2|s_3(t)|^\sigma + \hat{\varepsilon}_3 + \hat{\gamma}_3) sign(s_3(t)))
\end{cases}
\tag{42}
$$

Where, $\hat{\varepsilon}_i$ and $\hat{\gamma}_i$ $(i = 1, 2, 3.)$ are obtained by the adaptive update laws (25)

$$
\begin{cases}
D^\beta \hat{\varepsilon}_1 = |s_1| - k_2|\hat{\varepsilon}_1 - \varepsilon_1|^\sigma sign(\hat{\varepsilon}_1 - \varepsilon_1) \\
D^\beta \hat{\varepsilon}_2 = |s_2| - k_2|\hat{\varepsilon}_2 - \varepsilon_2|^\sigma sign(\hat{\varepsilon}_2 - \varepsilon_2) \\
D^\beta \hat{\varepsilon}_3 = |s_3| - k_2|\hat{\varepsilon}_3 - \varepsilon_3|^\sigma sign(\hat{\varepsilon}_3 - \varepsilon_3)
\end{cases}
$$

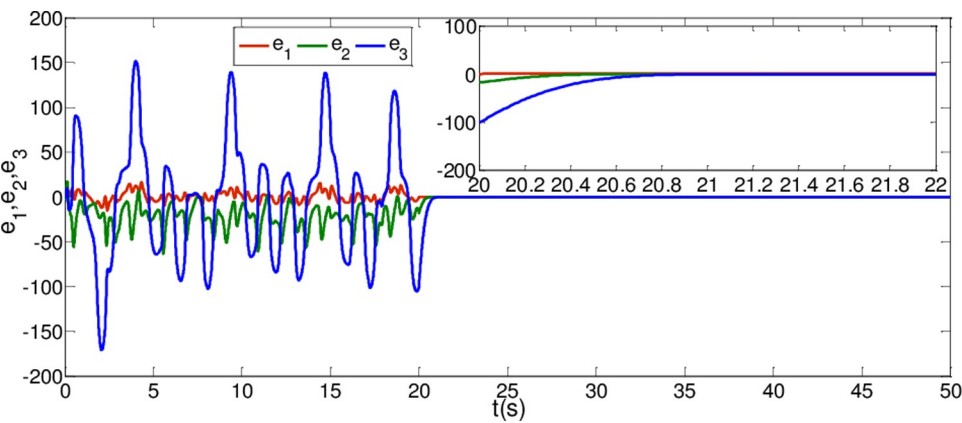

**Fig 5. State trajectories of the synchronization errors between master and slave system when α = 0.98 and β = 0.99.**

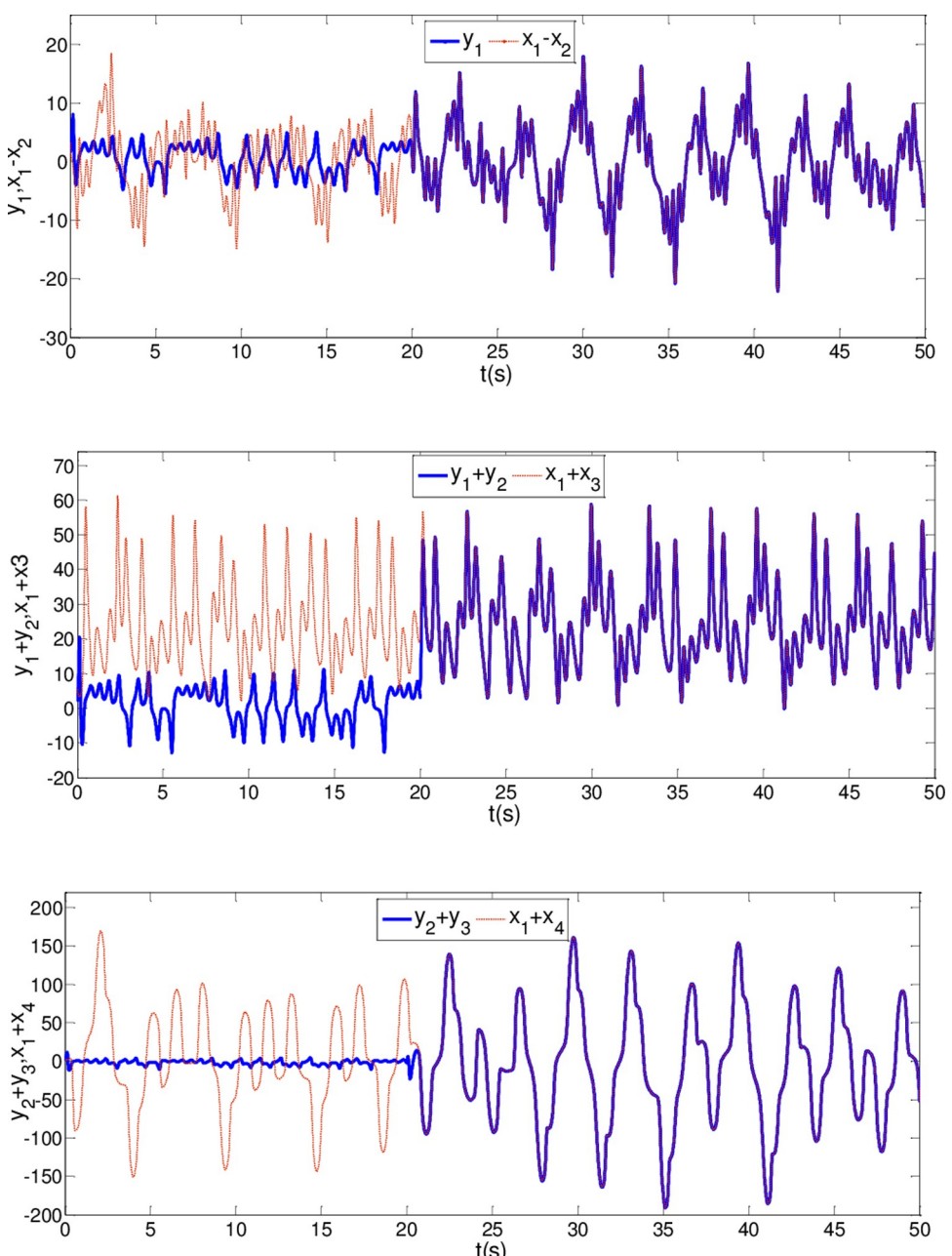

**Fig 6. State trajectories of master system and slave system when α = 0.98 and β = 0.99.**

$$\begin{cases} D^{\beta}\hat{\gamma}_1 = |s_1| - k_2|\hat{\gamma}_1 - \gamma_1|^{\sigma} sign(\hat{\gamma}_1 - \gamma_1) \\ D^{\beta}\hat{\gamma}_2 = |s_2| - k_2|\hat{\gamma}_2 - \gamma_2|^{\sigma} sign(\hat{\gamma}_2 - \gamma_2) \\ D^{\beta}\hat{\gamma}_3 = |s_3| - k_2|\hat{\gamma}_3 - \gamma_3|^{\sigma} sign(\hat{\gamma}_3 - \gamma_3) \end{cases}$$

In control scheme (42), The initial conditions of $\hat{\varepsilon}_i(0)(i = 1, 2, 3)$ and $\hat{\gamma}_i(0)(i = 1, 2, 3)$ are chosen as 0.5, the control parameters are chosen as $k_1 = k_2 = 10$, both $\delta$ and $\sigma$ are set equal to

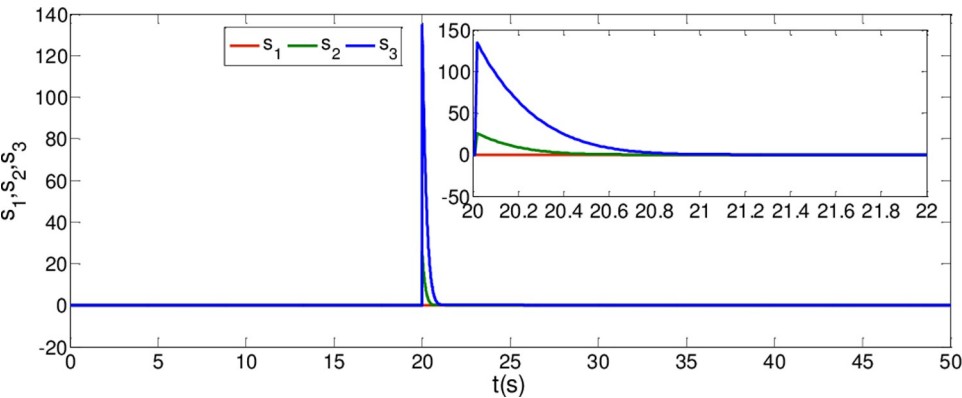

**Fig 7. Time response of the sliding surfaces.**

0.8. It's noted that the controller is performed at $t_0 = 20s$ and start from initial state $(x_1(20), x_2(20), x_3(20), x_4(20)) = (8.38, 5.89, 12.08, 89.39)$, $(y_1(20), y_2(20), y_3(20)) = (2.43, 0.75, -2.83)$, thus $e(20) = \varphi(y(20)) - \phi(x(20)) = (-0.06, -17.28, -99.85)$.

The synchronization errors of the master system (36) and slave system (37) are plotted in Fig 5. Obviously, the synchronization errors converge to zero rapidly, and the stabilized time is estimated within 1.00 s, which indicates that the global synchronization is successfully realized, as depicted in Fig 6. Furthermore, the corresponding time response of the sliding surface (13) is plotted in Fig 7.

According to Theorem 1, the states of the sliding mode dynamics system (14) will converge within a given time

$$T_1^* = t_0 + \left( \|e(t_0)\|_1^{\beta - \delta} \frac{\Gamma(1-\delta)\Gamma(1+\beta)}{k_1 \Gamma(1+\beta-\delta)} \right)^{\frac{1}{\beta}}$$

$$= 0 + (\|e(20)\|_1^{0.99-0.8})^{\frac{1}{0.99}} \times \left( \frac{\Gamma(1-0.8)\Gamma(1.99)}{10 \times \Gamma(1.99-0.8)} \right)^{\frac{1}{0.99}} = 1.2298$$

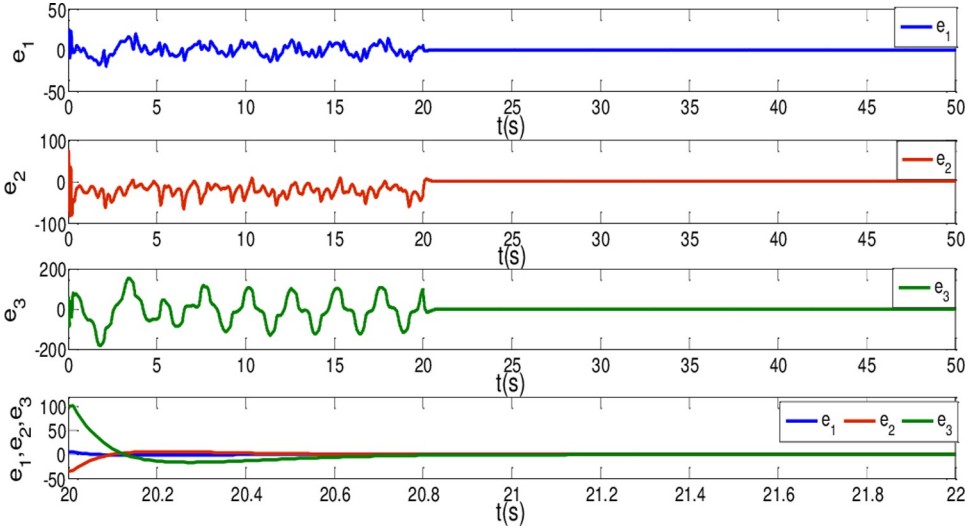

**Fig 8. State trajectories of the synchronization errors between master and slave system when $\alpha = \beta = 0.96$.**

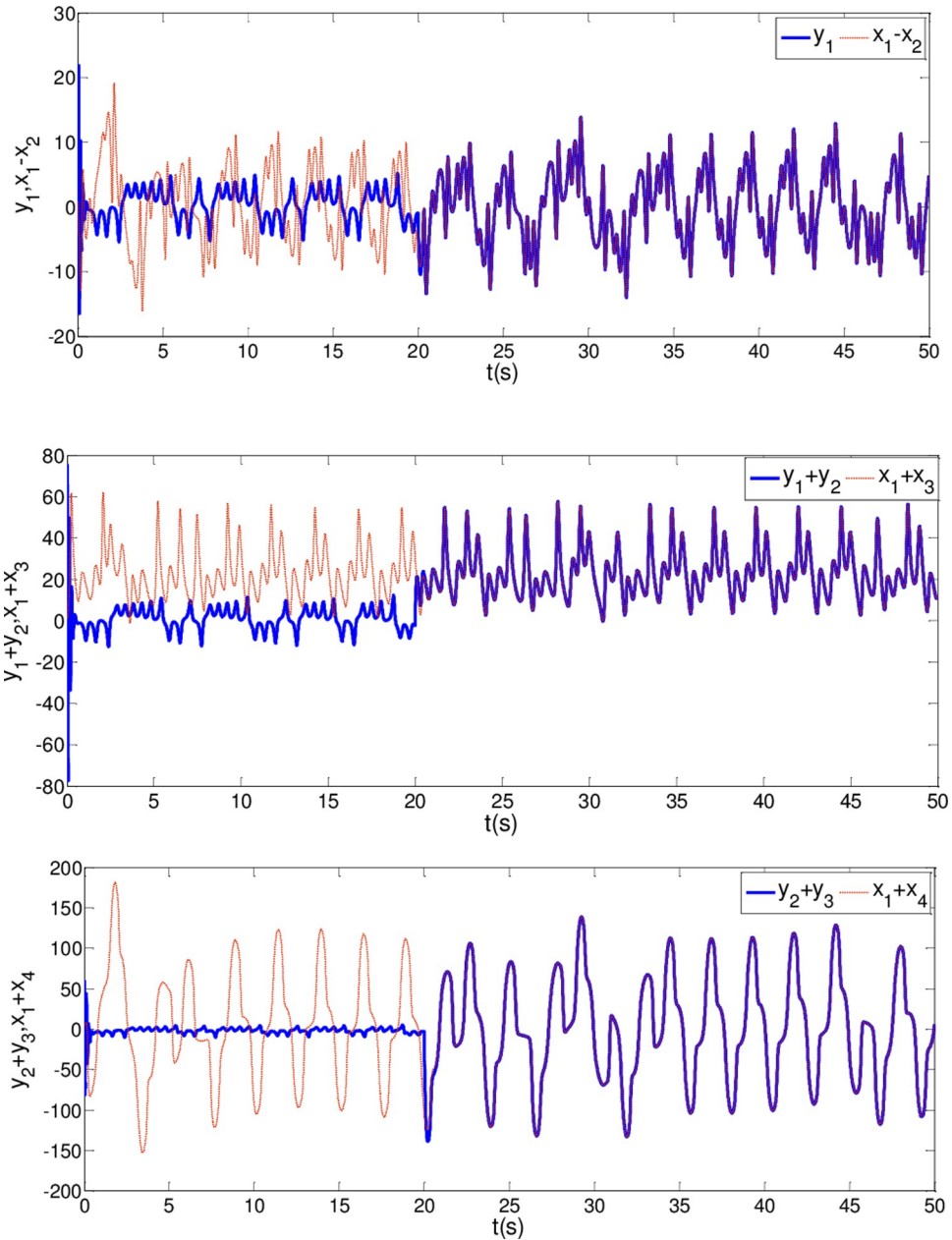

**Fig 9. State trajectories between master system and slave system when α = β = 0.96.**

In Fig 6, it is seen that the error system (12) can stabilize to zero with the reaching time of $t_1 \approx 0.8s$ and satisfy the estimated reaching time $t_1 \leq T_1^*$.

Besides, by (19) based on Definition 1 and Theorem 2, the estimated time is calculated as

$$T_2^* = t_0 + \left( \left( \frac{1}{2}s_i^2(0) + \frac{1}{2}\left( \frac{\hat{\varepsilon}_i(0)}{-\varepsilon_i} \right)^2 + \frac{1}{2}\left( \frac{\hat{\gamma}_i(0)}{-\gamma_i} \right)^2 \right)^{\beta - \frac{\sigma+1}{2}} \frac{\Gamma\left(1 - \frac{\sigma+1}{2}\right)\Gamma(1+\beta)}{2^{\frac{\sigma+1}{2}}k_2\Gamma\left(1+\beta-\frac{\sigma+1}{2}\right)} \right) = 1.222$$

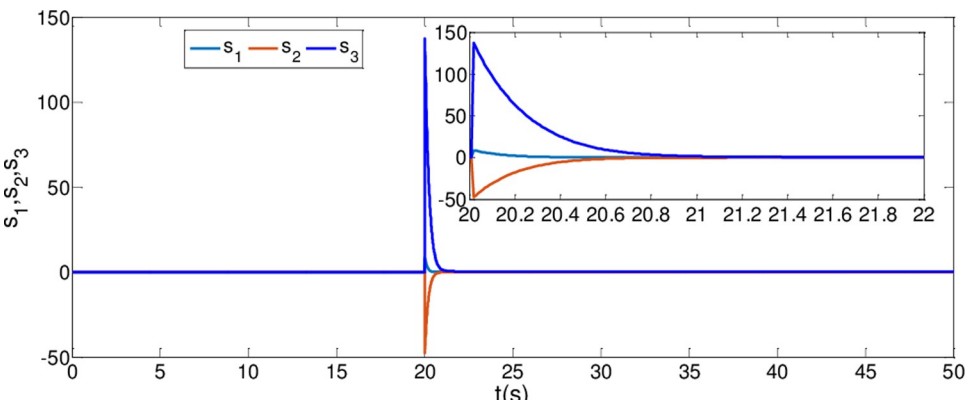

**Fig 10. Time response of the sliding surfaces.**

From Fig 7, it is obvious that the simulation time $t_2 \approx 0.8s$ satisfies the estimated time $t_2 \leq T_2^*$.

## 4.2. Commensurate order

Assume that the commensurate order $\alpha = \beta = 0.96$ and the initial values of the systems (36) and (37) are set as $(x_1, x_2, x_3, x_4) = (3,1,-2,1)$, $(y_1, y_2, y_3) = (1,2,3)$. And the control signals $U(t) = U'(t)+U''(t)$, compensation controllers $U''_i(t) = J_\varphi^{-1} J_\phi (D^\beta x - D^\alpha x) = 0$, $i = 1,2,3$. Then, adaptive controllers $U'_i(t)$ are determined by the form of (42). The other parameter values are the same as Case I. It is noteworthy that the controller is implemented at $t_0 = 20s$. The results are depicted in Figs 8–10. As shown in Figs 8 and 9, the generalized synchronization is successful accomplished and the synchronization errors tend to zero within 1.0s. From Fig 10, it is clear that the sliding surfaces converge to zero within 1.0s. The calculation method of convergence time is similar to Theorem 1 and 2, the upper bound of those can be estimated as $T_1^* =$

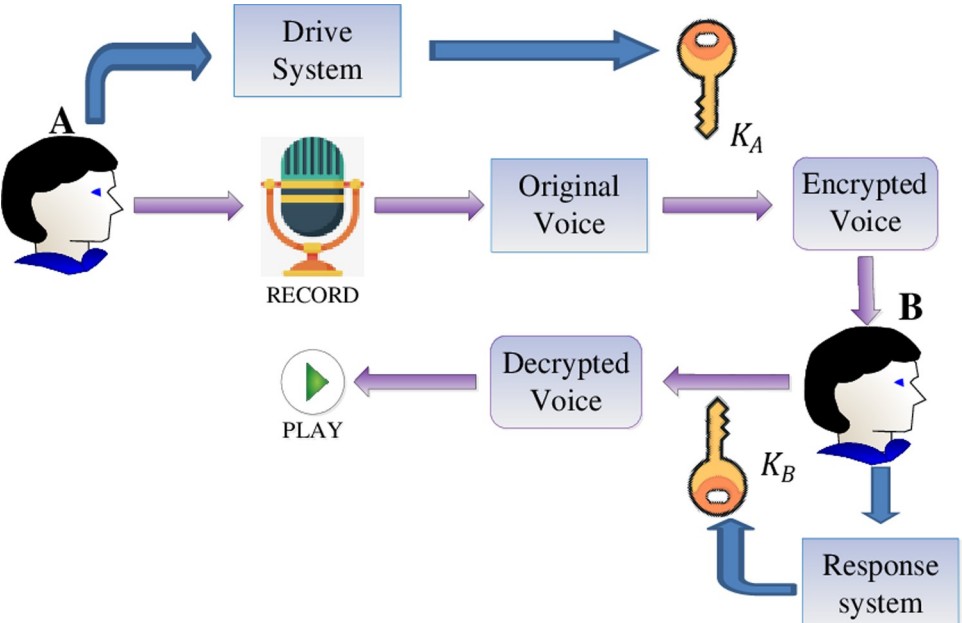

**Fig 11. The overall diagram of speech secure communication.**

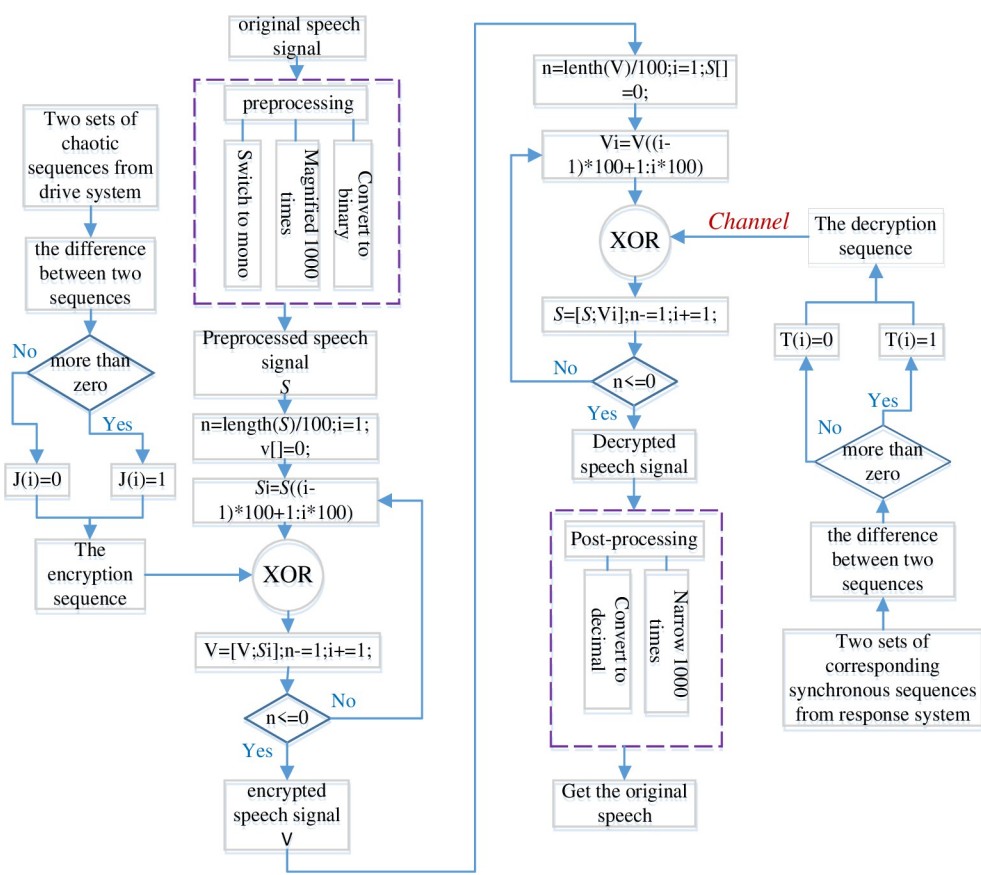

**Fig 12. Encryption and decryption scheme of the speech signal based on synchronization criterion.**

0.9141, $T_2^* = 0.9438$ respectively. The results verify the presented control strategy can synchronize two different dimensional fractional order chaotic systems within finite-time.

**Remark 8** For the proposed synchronization approach, the error state trajectories with both cases will converge to zero within 1.0s according to Figs 5 and 10. But the existing method in reference [18], the synchronization errors converge to zero at the reaching time t ≥1.6s. Therefore, the convergence rate of synchronization errors for fractional order chaotic systems is faster than that of the existent approach in [18], which implies the superiority and effectiveness of the presented scheme in our study. In addition, integer order dynamical system is also appropriate.

**Remark 9** According to **Remark 8**, it is obvious that the novelty and superiority of the model is highlighted by the proposed approach. The finite-time generalized synchronization can be applied to both commensurate and incommensurate systems. Moreover, the fact that it acquires the coexistence of several kinds of synchronization types, as see in [12], the coexistence of three different synchronization types, that is, identical synchronization (IS), anti-phase synchronization (AS), and inverse full state hybrid projective synchronization (IFSHPS). But the fact that the proposed approach is more general than that in [12], since it guarantees the combination of only three specific synchronization types. Furthermore, the proposed synchronization approach gives a deeper insight into the synchronization phenomena between fractional order chaotic systems.

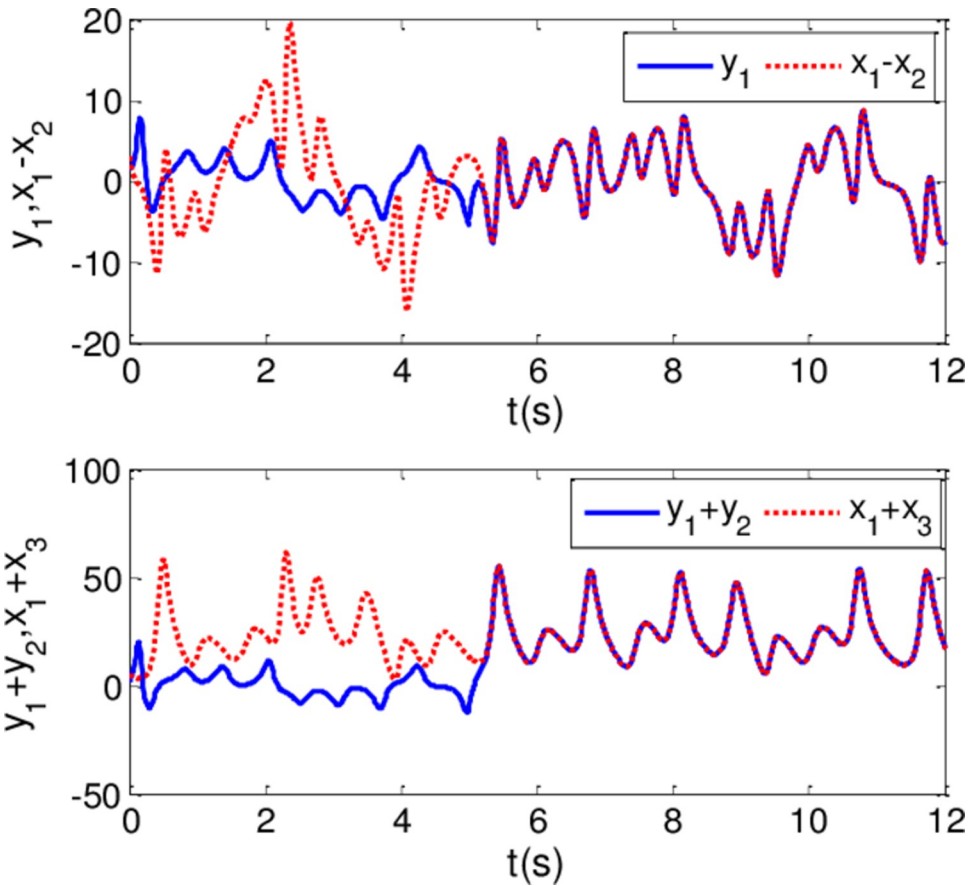

**Fig 13. Synchronous trajectories of drive system and response system when α = 0.98 and β = 0.99.**

## 5. Applications in speech secure communication

In this section, due to the technology of finite-time synchronization and uncertainties of the synchronous system can enhance the security of communication. A new speech cryptosystem is proposed to send or share voice messages privately according to generalized finite-time synchronization criterion of non-identical fractional order chaotic drive system and response system improving the level of security. Before signal transmission, based on synchronization theory among fractional-order chaotic systems, the generalized synchronization errors of the systems (36) and (37) will converge to zero under the given control inputs (42) and a time $t > t_1 + t_2$. Then, the sender A records an audio message and generates him/her own key of chaotic sequence $K_A$. The receiver B obtains the secret key of chaotic sequence $K_B$ by means of the proposed synchronization criterion and then decrypts the original speech signal. It is noted that encryption-decryption keys $K_A$, $K_B$ are the same. Furthermore, the overall diagram of speech encryption–decryption process is depicted in Fig 11.

### 5.1. Description of encryption and decryption scheme

Now the speech encryption and decryption algorithm is designed based on generalized synchronization of fractional order chaotic system. Chaotic sequences can be generated by using the systems (36) and (37), and the arbitrary two functions of $\phi(x)$ from the systems (36) are chosen to generate the key of encryption algorithm. The original sound is then encrypted by

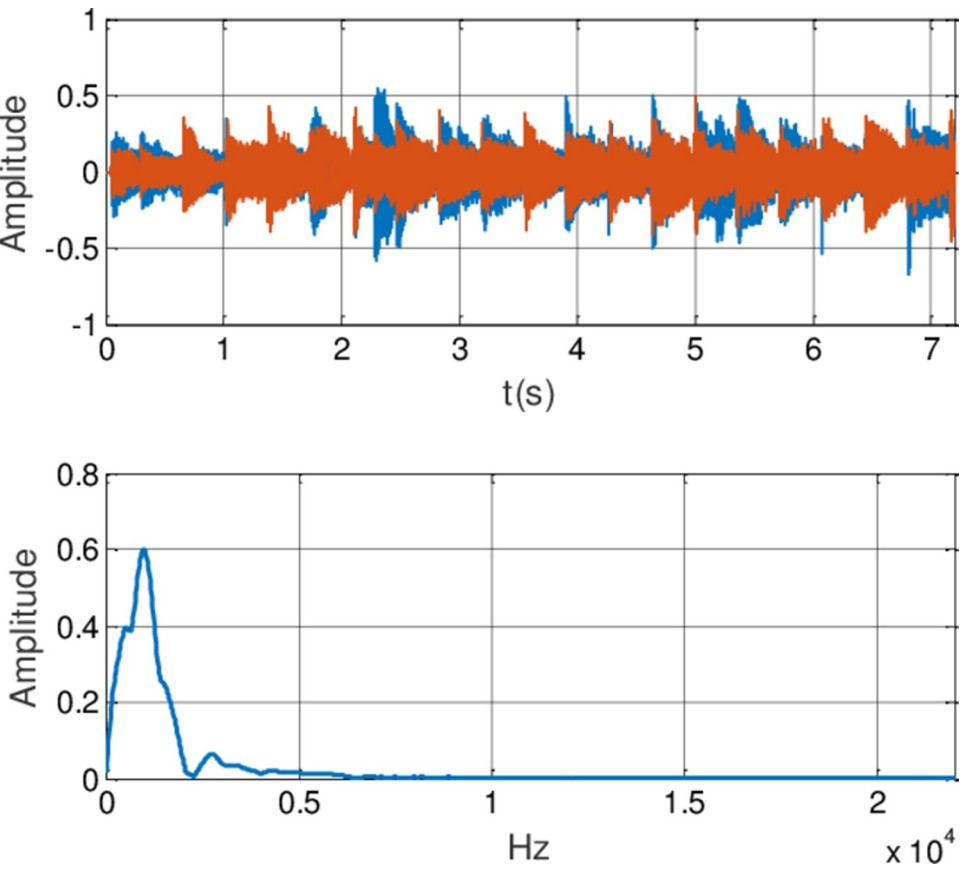

**Fig 14. The original speech signal and FFT spectrum.**

employing a simple XOR operation with the chaotic sequence generated. The decryption algorithm is basically similar to the encryption algorithm. As shown in Fig 12, the complete algorithm can be described in the following steps:

*Step 1*: The preliminary speech signal preprocessing. Selecting any channel signal of the recorded two-channel speech signals. The chosen signal will realize the conversion from decimal to binary. And its amplitude will be magnified 1000 times.

*Step 2*: The encryption sequence generated. The arbitrary two functions of $\phi(x)$ from the systems (36) are chosen to generate the chaotic sequence $a_1$, $b_1$. Then there will get the difference of a hundred numbers from $a_1$ and $b_1$. If $a_1(i) - b_1(i) > 0$, $i = 1,2,3 \cdots 100$, the sequence of encryption J$(i) = 1$, In contrast, J$(i) = 0$, that is to say, the key of encryption $K_A$ has been generated.

*Step 3*: Encryption. The sequence of encryption J$(i)$ will be transformed into binary number and then be encrypted by using a simple XOR operation.

*Step 4*: Decryption. The corresponding two functions of $\varphi(y)$ from the systems (37) are obtained to generate the tracking sequence $a_2$, $b_2$ in terms of synchronization criterion. Additionally, the decryption algorithm is the same as the encryption algorithm. Then the sequence of decryption T$(i)$ could be generated. Likewise, the corresponding XOR operation will also been performed. The decrypted speech signal can be acquired.

*Step 5*: Decrypted speech signal post-processing. The decrypted speech signal will realize the conversion from binary to decimal, and its amplitude will be reduced by 1000 times.

*Step 6*: Get the original speech.

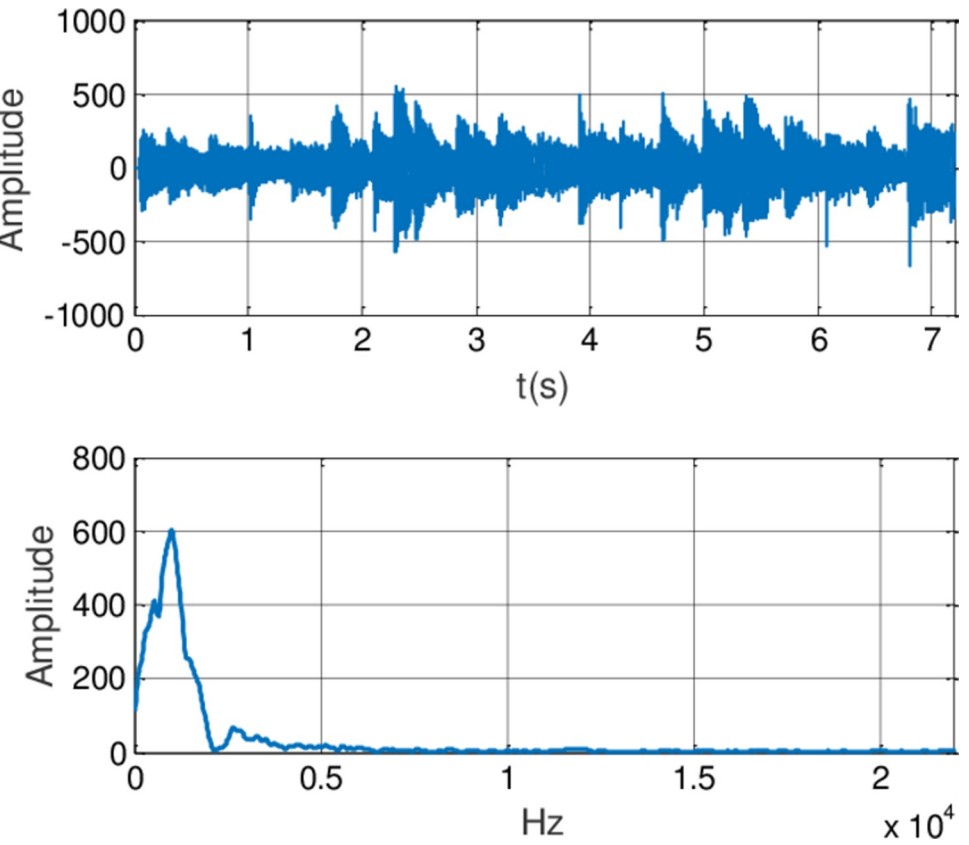

**Fig 15. Original speech signal preprocessing and FFT spectrum.**

## 5.2. Demonstration of the experimental results

In this experiment, the generalized finite-time synchronization between the systems (36) and (37) is implemented in Case I, while the controller is applied at $t_0 = 5s$. Synchronous trajectories are depicted in Fig 13. Further, Assume that A records a speech message and wishes to send it to B secretly. Both A and B should be agreeing on a time $t > t_0$. The message has been saved by A as the audio format of s.mp3.The original speech signal and corresponding FFT spectrum is displayed in Fig 14. It has 7.2 seconds long with 44100 samples. For encryption, the preprocessing for original speech signal is completed in advance. As shown in Fig 15, the original speech signal preprocessing and FFT spectrum are represented graphically. From Fig 16, the arbitrary 40 points chosen from the original speech are converted to the binary bits. According to encryption scheme in the previous section, encrypted speech signal and its corresponding FFT spectrum are illustrated in Fig 17. It is clear that the speech signal is entirely covered by the chaotic secret key sequence, and the original profile could not be seen at all. For decryption, the decrypted speech signal and its corresponding FFT spectrum are depicted in Fig 18. Finally, B will obtain an original speech message without any loss of information, because the decrypted audio signal is fully restored. Therefore, B can play decrypt speech and hear an original voice. It is well known that the secret key sequence generated by the drive system is utilized to encrypt the speech signal and that generated by the response system operates in recovering the encryption signal.

**Remark 9** Notice that from the aforementioned result, spectrogram analysis is a random test tool that divides the signal in the time domain into slots to calculate Fast Fourier

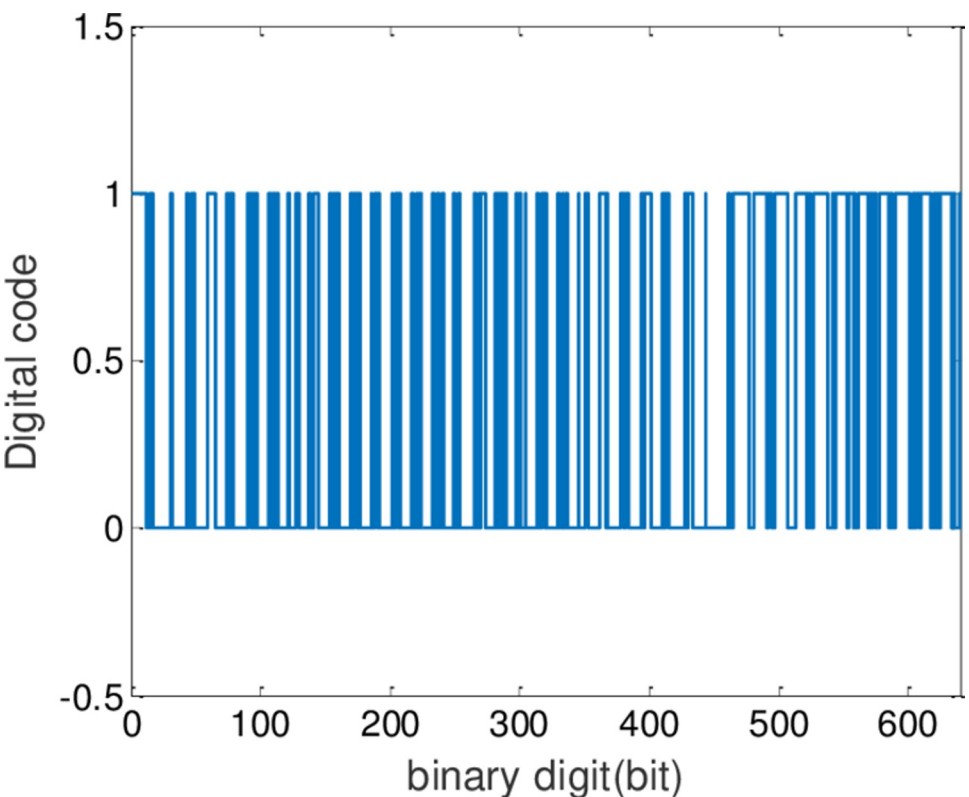

**Fig 16. The binary number of the original speech signal.**

Transform (FFT) for each slot. The magnitude square of FFT is plotted versus each slot to indicate the energy of sound. There are 44100 samples of the input speech signal with7.2 seconds long. Then, the spectrogram of original and encrypted signals is made. The energy of sound appears to be small as a result of low speech signal's amplitude, so its amplitude will be magnified 1000 times. For encrypting the speech file, spectrogram appears random which indicates the randomness distribution of sound components' energy. In the decryption process, the waveforms of the decrypted speech signal are shown by the decryption key. Encryption and decryption scheme of the speech signal based on synchronization criterion is straightforward with low-level hardware complexity. Furthermore, the proposed method also effectively enhances the security of signal transmission. Image encryption is one of the most significant and common applications of synchronization between fractional order chaotic systems. A future direction of investigation is to recast the methodology adopted in this paper to the image encryption.

### 5.3. Security analysis

In the proposed synchronization criterion, the fractional order chaotic systems (36) and (37) are used to generate the secret key of encryption and decryption sequences $K_A$, $K_B$. Due to some secret elements of fractional-order chaotic systems, such as the parameters and initial conditions of the system, fractional orders α and β and the convergence time, will add the total number of different secret keys directly. Hence, the new algorithm has a large enough key space to resist brute-force attacks. The key sequences generated are totally uncorrelated and random. When such sequences are utilized, experimental results have revealed that the

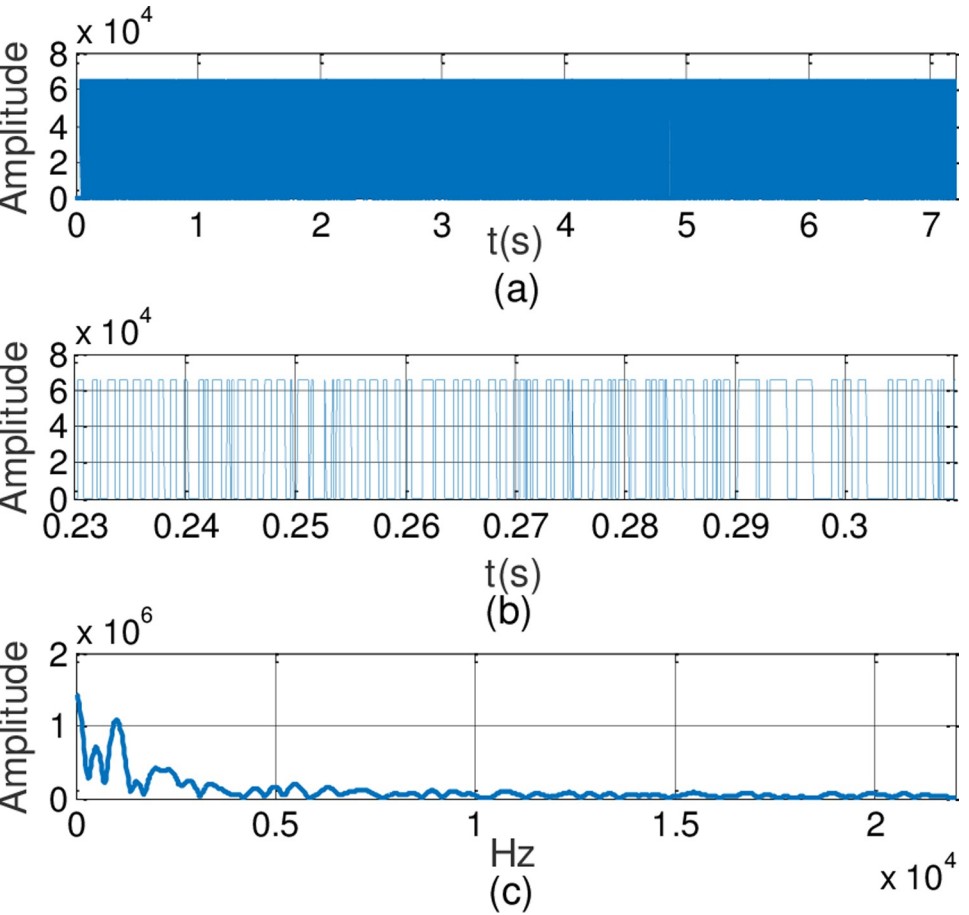

**Fig 17. Encrypted speech signal and FFT spectrum.**

proposed algorithm possesses some secret elements of traditional cryptography, such as complex chaotic behavior, time-varying nonlinearity, disturbances and model uncertainties. Therefore, the reverse recover of the original speech message is totally hopeless except by the receiver. Meanwhile, the demonstration and analysis of the speech cryptosystem based the fractional order dynamical systems have shown that, the proposed encryption and decryption scheme is more secure and appropriate for sending and receiving messages secretly. Furthermore, these have basically addressed all existing security disadvantages with respect to chaos based audio encryption methods and have provided a new idea for the ever-increasing practical applications.

## 6. Conclusion

This paper is concerned with the generalized finite-time synchronization between two non-identical fractional order chaotic (or hyper-chaotic) systems based on adaptive sliding mode controller and its application in secure communication. First, the definition of generalized finite-time synchronization is given. Second, a novel fractional order integral sliding surface is presented and its finite-time convergence theorem is analytically proved. Then, according to the fractional Lyapunov stability theory, a robust controller with adaptive update laws is proposed to ensure the occurrence of the sliding motion. Meanwhile, its finite-time stability condition is derived by considering model uncertainties and external disturbances. The results of

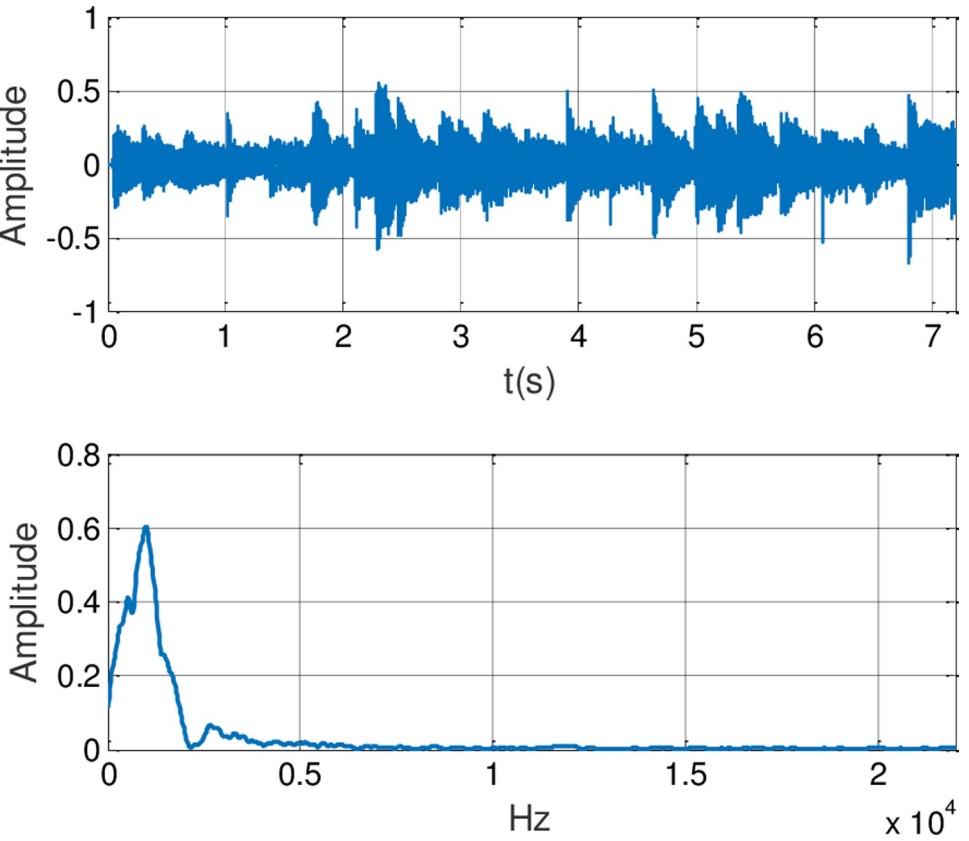

**Fig 18. Decrypted speech signal and FFT spectrum.**

theoretical analysis show that the finite-time stability and the robustness of the proposed control scheme are mathematically proved, and the upper bound of the convergence time is explicitly evaluated. Finally, numerical simulations illustrate the effectiveness and robustness of the presented approach, which are in good agreement with the results of theoretical analysis. What's more, in order to demonstrate the practical effect of generalized synchronization with application to the speech secure communication, a novel sound encryption mechanism is proposed and a successful case is given to show the applicability of the proposed theories. It is worthwhile to note that the proposed synchronization approach not only can be extended to a wide range of nonlinear fractional-order chaotic systems and time-delayed chaotic systems, but also can be further applied to create a new encryption mechanism or a new way guaranteeing information safety. A future direction of investigation is to recast the methodology adopted in this paper to the image encryption.

## Supporting information

**S1 File.**
(ZIP)

## Author Contributions

**Conceptualization:** Jianxiang Yang.

**Formal analysis:** Jian Cen, Wei He.

**Funding acquisition:** Jian Cen.

**Investigation:** Jianbin Xiong.

**Methodology:** Jianxiang Yang.

**Software:** Jianxiang Yang, Jianbin Xiong, Wei He.

**Writing – original draft:** Jianxiang Yang.

**Writing – review & editing:** Jianxiang Yang, Wei He.

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
