## [Decision Letter · Decision Letter 0]

3 Nov 2021

PONE-D-21-24308Finite-time generalized synchronization of non-identical fractional order chaotic systems and its application in speech secure communicationPLOS ONE

Dear Dr. Yang,

Thank you for submitting your manuscript to PLOS ONE. After careful consideration, we feel that it has merit but does not fully meet PLOS ONE’s publication criteria as it currently stands. Therefore, we invite you to submit a revised version of the manuscript that addresses the points raised during the review process.

We look forward to receiving your revised manuscript.

Kind regards,

Academic Editor

PLOS ONE

“This study was supported in part by the National Natural Science Foundation of China under Grant no. 62073090, 61473331, in part by the Guangzhou Key Laboratory of Intelligent Building Equipment Information Integration and Control under Grant no. 202002010003, in part by the Natural Science Foundation of Guangdong Province of China under Grant no.2019A1515010700, in part by the Key (natural) Project of Guangdong Provincial under Grant no. 2019KZDXM020,2019KZDZX1004, 2019KZDZX1042, in part by the Introduction of Talents Project of Guangdong Polytechnic Normal University of China under Grant no. 991641277, 991512203, in part by Guangdong Climbing Project no. pdjh2020b0345, in part by Special projects in key areas of ordinary colleges and universities in Guangdong Province no. 2020ZDZX2014,Intelligent Agricultural Engineering Technology Research Centre of Guangdong University Grant no.ZHNY1905, and Guangzhou Key Laboratory of Intelligent Building Equipment Information Integration and Control.”

“This study was supported in part by the National Natural Science Foundation of China under Grant no. 62073090, 61473331, in part by the Guangzhou Key Laboratory of Intelligent Building Equipment Information Integration and Control under Grant no. 202002010003, in part by the Natural Science Foundation of Guangdong Province of China under Grant no.2019A1515010700, in part by the Key (natural) Project of Guangdong Provincial under Grant no. 2019KZDXM020,2019KZDZX1004, 2019KZDZX1042, in part by the Introduction of Talents Project of Guangdong Polytechnic Normal University of China under Grant no. 991641277, 991512203, in part by Guangdong Climbing Project no. pdjh2020b0345, in part by Special projects in key areas of ordinary colleges and universities in Guangdong Province no. 2020ZDZX2014,Intelligent Agricultural Engineering Technology Research Centre of Guangdong University Grant no.ZHNY1905, and Guangzhou Key Laboratory of Intelligent Building Equipment Information Integration and Control.”

“NO authors have competing interests”

Additional Editor Comments:

Based on the two reviewers' suggestions, the paper needs to be revised largely from the technique to the writing.

Reviewers' comments:

Reviewer's Responses to Questions

**Comments to the Author**

1. Is the manuscript technically sound, and do the data support the conclusions?

Reviewer #1: Yes

Reviewer #2: Yes

2. Has the statistical analysis been performed appropriately and rigorously? 

Reviewer #1: Yes

Reviewer #2: N/A

3. Have the authors made all data underlying the findings in their manuscript fully available?

Reviewer #1: Yes

Reviewer #2: No

4. Is the manuscript presented in an intelligible fashion and written in standard English?

Reviewer #1: Yes

Reviewer #2: Yes

5. Review Comments to the Author

Reviewer #1: In my opinion, the article is innovative and suitable for publication in this journal after the following  revisions can be finished.

1 This article needs further careful examination to avoid language errors that should not occur, for example at the beginning of the Introduction, there are two Chaos Chaos; Title of Chapter II：“2. Basic definition and lemma”I think “definitions”should be more suitable.

2.The reference format of the article is very chaotic, and the author needs to revise it uniformly according to the requirements of the journal.

3.The relationship between theorem 1 and theorem 2 is not well explained, which is easy to cause confusion in understanding. Please explain in detail the relationship between theorem 1 and theorem 2.

4.Figure 1-4 only list the behavior evolution of system state and output, without further detailed comparative analysis. Please give a detailed comparative analysis of status and output signals.

Reviewer #2: The problem of the finite-time generalized synchronization for non-identical fractional order chaotic (or hyper-chaotic) systems by a designing adaptive sliding mode controller and its application to secure communication has been focused on in this manuscript. And some simulation results have been provided to demonstrate the effectiveness and robustness of the presented approach. However, the following comments need to be considered in the revision and require major revisions.

(1) What are the advantages of the fractional order system considered in this paper over the integer order system? What are the differences and difficulties in handling?

(2) The contributions of this paper need to be improved.

(3) The background in the introduction part is not enough, please refer to the following reference:

IEEE Transactions on Neural Networks and Learning Systems, DOI: 10.1109/TNNLS.2019.2952410.

(4) In the Definitions 1-2, Lemmas 2, 3, 5 in the paper, they require \\alpha \\in (0,1), while Lemma 1 requires \\alpha \\in (0,1], which makes the definitions 1-2, Lemmas 2, 3, 5 no longer hold when Lemma 1 holds. Please explain this contradiction issue in detail.

(5) The symbol “\\varepsilon_{i}” given in Lemma 4 is not reflected in the following inequality (6), and the symbol \\xi in (6) does not give a corresponding definition. In addition, there are many symbols in the full paper that appear for the first time without giving corresponding definitions, such as the symbol \\tau, the symbol before the function f(t) in definition 1, etc.

(6) The authors mentioned in Remark 4 that “designing an appropriate control law ′() for any different dimensional systems (9) and (10)”? Is this designed control law applicable to any different dimensional systems?

(7) What is the prior knowledge of the upper bound of the system mentioned by the authors in Remark 6? In the control scheme proposed in this paper, why is it not necessary to give a priori knowledge of the upper bound? What method was used to deal with it?

(8) The expression of the finite time T calculated in the paper contains the initial values of some variables. How are these initial values determined?

6. PLOS authors have the option to publish the peer review history of their article (what does this mean?). If published, this will include your full peer review and any attached files.

Reviewer #1: No

Reviewer #2: No

---

## [Author Response · Author response to Decision Letter 0]

2 Dec 2021

We thank the innominate reviewers for their valuable comments and suggestions in improving the quality of the manuscript. We have modified the new manuscript following your advice. Thank you for your suggestion again.

---

## [Decision Letter · Decision Letter 1]

11 Jan 2022

Finite-time generalized synchronization of non-identical fractional order chaotic systems and its application in speech secure communication

PONE-D-21-24308R1

Dear Dr. Yang,

We’re pleased to inform you that your manuscript has been judged scientifically suitable for publication and will be formally accepted for publication once it meets all outstanding technical requirements.

Kind regards,

Academic Editor

PLOS ONE

Additional Editor Comments (optional):

Based on the two reviewers' comments, the paper can be accepted for publication now.

Reviewers' comments:

Reviewer's Responses to Questions

**Comments to the Author**

1. If the authors have adequately addressed your comments raised in a previous round of review and you feel that this manuscript is now acceptable for publication, you may indicate that here to bypass the “Comments to the Author” section, enter your conflict of interest statement in the “Confidential to Editor” section, and submit your "Accept" recommendation.

Reviewer #1: All comments have been addressed

Reviewer #2: All comments have been addressed

2. Is the manuscript technically sound, and do the data support the conclusions?

Reviewer #1: Yes

Reviewer #2: (No Response)

3. Has the statistical analysis been performed appropriately and rigorously? 

Reviewer #1: Yes

Reviewer #2: (No Response)

4. Have the authors made all data underlying the findings in their manuscript fully available?

Reviewer #1: Yes

Reviewer #2: (No Response)

5. Is the manuscript presented in an intelligible fashion and written in standard English?

Reviewer #1: Yes

Reviewer #2: (No Response)

6. Review Comments to the Author

Reviewer #1: Authors have made good modifications and basically answered my questions. In my opinion, it's acceptable.

Reviewer #2: (No Response)

7. PLOS authors have the option to publish the peer review history of their article (what does this mean?). If published, this will include your full peer review and any attached files.

Reviewer #1: No

Reviewer #2: No

---

## [Editor Report · Acceptance letter]

14 Mar 2022

PONE-D-21-24308R1 

Finite-time generalized synchronization of non-identical fractional order chaotic systems and its application in speech secure communication 

Dear Dr. Yang:

I'm pleased to inform you that your manuscript has been deemed suitable for publication in PLOS ONE. Congratulations! Your manuscript is now with our production department. 

Kind regards, 

on behalf of

Prof. Yanzheng Zhu 

Academic Editor

PLOS ONE